# A generalized cortical activity pattern at internally generated mental context boundaries during unguided narrative recall

Hongmi Lee*, Janice Chen

Department of Psychological and Brain Sciences, Johns Hopkins University, Baltimore, United States

**Abstract** Current theory and empirical studies suggest that humans segment continuous experiences into events based on the mismatch between predicted and actual sensory inputs; detection of these 'event boundaries' evokes transient neural responses. However, boundaries can also occur at transitions between internal mental states, without relevant external input changes. To what extent do such 'internal boundaries' share neural response properties with externally driven boundaries? We conducted an fMRI experiment where subjects watched a series of short movies and then verbally recalled the movies, unprompted, in the order of their choosing. During recall, transitions between movies thus constituted major boundaries between internal mental contexts, generated purely by subjects' unguided thoughts. Following the offset of each recalled movie, we observed stereotyped spatial activation patterns in the default mode network, especially the posterior medial cortex, consistent across different movie contents and even across the different tasks of movie watching and recall. Surprisingly, the between-movie boundary patterns did not resemble patterns at boundaries between events within a movie. Thus, major transitions between mental contexts elicit neural phenomena shared across internal and external modes and distinct from within-context event boundary detection, potentially reflecting a cognitive state related to the flushing and reconfiguration of situation models.

*For correspondence:
hongmi.lee@jhu.edu

## Editor's evaluation

This paper provides convincing evidence that internally generated event boundaries occurring at abrupt shifts in mental state evoke similar neural responses as those triggered by a change in sensory input. Given that much past work has linked the detection of event boundaries to the discrepancy between prediction and input, these new findings are significant and anticipated to spur much future research on event boundaries in the absence of external change. This innovative and methodologically rigorous study will be of interest to cognitive neuroscientists working on topics broadly related to memory, event segmentation, and mental context.

## Introduction

Humans perceive and remember continuous experiences as discrete events (*Brunec et al., 2018*; *Clewett et al., 2019*; *Shin and DuBrow, 2021*; *Zacks, 2020*). Studies of event segmentation have shown that when participants attend to external information (e.g., watch a video), (1) boundaries between events are detected when mismatches arise between predicted and actual sensory input (*Zacks et al., 2007*; *Zacks et al., 2011*), and (2) boundary detection evokes transient neural responses

**Figure 1.** Experimental procedures and univariate responses. (**A**) In the encoding phase, subjects watched 10 short movies approximately 2–8 min long. Each movie started with a 6 s title scene. In the free spoken recall phase, subjects verbally recounted each movie plot in as much detail as possible regardless of the order of presentation. After recalling one movie, subjects spontaneously proceeded to the next movie, and the transitions between movies were considered as internally driven boundaries. Red arrows indicate the boundaries (onsets and offsets) between watched or recalled movies. Black arrows indicate the non-boundary moments (middle) of each watched or recalled movie. (**B**) Whole-brain maps of unthresholded mean activation (blood oxygen level-dependent [BOLD] signals z-scored across all volumes within a scanning run) following between-movie boundaries during recall (4.5–19.5 s from the offset of each movie). Blue areas indicate regions with lower-than-average activation, where the average activation of a scanning run was z = 0. Likewise, red areas indicate regions with higher-than-average activation. White outlines indicate areas that showed significantly lower or higher activation following between-movie boundaries compared to non-boundary periods (false discovery rate-corrected q < 0.05; minimum surface area = 16 mm²). The non-boundary periods were defined as the middle 15 s of each recalled movie, shifted forward by 4.5 s. Changes in whole-brain univariate responses across time around the boundaries are shown in *Figure 1—video 1* (recall phase) and *Figure 1—video 2* (encoding phase).

The online version of this article includes the following video, source data, and figure supplement(s) for figure 1:

**Figure supplement 1.** Mean activation time courses around between-movie boundaries.

**Figure supplement 1—source data 1.** Source data for *Figure 1—figure supplement 1*.

**Figure 1—video 1.** Changes in univariate activation at between-movie boundaries during recall (video).
https://elifesciences.org/articles/73693/figures#fig1video1

**Figure 1—video 2.** Changes in univariate activation at between-movie boundaries during encoding (video).
https://elifesciences.org/articles/73693/figures#fig1video2

in a consistent set of brain areas (*Reagh et al., 2020*; *Speer et al., 2007*; *Zacks et al., 2001*). Among these areas is the default mode network (DMN; *Buckner and DiNicola, 2019*) proposed to be involved in representing complex mental models of events (*Ranganath and Ritchey, 2012*; *Ritchey and Cooper, 2020*). However, a substantial portion of human cognition is internally driven (*Hasselmo, 1995*; *Honey et al., 2018*), and such spontaneous production of thoughts and actions is also punctuated by mental context transitions (*Christoff et al., 2016*; *Mildner and Tamir, 2019*; *Smallwood and Schooler, 2015*; *Tseng and Poppenk, 2020*). What manner of brain activity marks boundaries between mental contexts when they are internally generated? Are the brain responses at internal boundaries similar to those at external boundaries?

Here, we used naturalistic movie viewing and free spoken recall with fMRI to characterize neural activity at boundaries between internally generated mental contexts (*Figure 1A*). Subjects watched 10 short movies (encoding phase), then verbally recounted the movies in any order, in their own words (recall phase). The transitions between recalled movies were determined purely by subjects' internal mentation; no external cues prompted the recall onset or offset of each movie. Moreover, the unguided spoken recall allowed us to identify the exact moments of context transitions and explicitly track shifts in the contents of thoughts (*Chen et al., 2017*; *Sripada and Taxali, 2020*), which was not possible in prior studies using silent rest (*Karapanagiotidis et al., 2020*; *Tseng and Poppenk, 2020*). At these internal boundaries between recalled movies, we observed transient, highly generalizable and fine-grained activation patterns throughout the DMN, consistent across diverse movie contents and similar to those at external between-movie boundaries during encoding. Moreover, these between-movie boundary patterns were not merely stronger versions of within-movie 'event boundaries,' but instead manifested as a distinct type of neural transition. We propose that these cortical patterns reflect a cognitive state related to the major flushing and reconfiguration of mental context (*DuBrow et al., 2017*; *Manning et al., 2016*).

## Results

We first examined whether internally driven boundaries evoke changes in blood oxygen level-dependent (BOLD) signals during recall. We observed transient changes in activation at the boundaries between recalled movies in widespread cortical regions (*Figure 1—video 1*; see *Figure 1—figure supplement 1* for activation time courses). A whole-brain analysis with multiple comparisons correction revealed that the mean activation of boundary periods (15 s following the offset of each movie) was generally lower than that of non-boundary periods (middle 15 s within each movie) in multiple areas, including the motor, auditory, and inferior parietal cortices, although a smaller number of regions showed higher activation during non-boundary periods (*Figure 1B*).

Next, we tested whether there were neural activation patterns specific to internally driven boundaries and consistent across different movies. We performed a whole-brain pattern similarity analysis on the recall data to identify regions where (1) boundary period activation patterns were positively correlated across different recalled movies (*Figure 2A*, blue arrow *a* > 0), and (2) this correlation was higher at boundaries than at non-boundaries (*Figure 2A*, blue arrows *a* > *b*). We observed a consistent boundary pattern, that is, whenever participants transitioned from talking about one movie to the next, in several cortical parcels (*Schaefer et al., 2018*), including the DMN and auditory/motor areas (*Figure 2B*). Thus, the boundary patterns within the recall phase were likely to be driven by both shared low-level sensory/motor factors (e.g., breaks in recall speech generation) as well as cognitive states (e.g., memory retrieval) at recall boundaries. No cortical parcel showed significantly negative correlations between boundary patterns or greater correlations in the non-boundary compared to boundary conditions.

To what extent is the internally driven boundary pattern, measured during recall, similar to patterns observed at boundaries during encoding? To test this, we again computed between-movie pattern similarity for all cortical parcels in the brain, but now across the encoding and recall phases (*Figure 2A*, red arrows). We found that DMN areas showed a consistent boundary pattern across task phases (encoding and recall) and across movies (*Figure 2C*). Again, no cortical parcel showed negative correlations between boundary patterns or greater correlations in the non-boundary condition. Among the DMN areas, the posterior medial cortex (PMC) showed the most consistent boundary patterns; thus, we next examined the phenomenon in more detail specifically in PMC. *Figure 3A and C* visualize the high and consistently positive correlations of PMC boundary patterns across different movies both within the recall phase (recall offset vs. recall offset, $t(14) = 11.82$, p<0.001, Cohen's $d_z$ = 3.05, 95% confidence interval (CI) = [0.28,0.41]) and even between experimental phases (recall offset vs. encoding offset, $t(14) = 14.54$, p<0.001, Cohen's $d_z$ = 3.75, 95% CI = [0.28,0.38]). No such correlation was present between non-boundary patterns ($t(14)$s < 1, $p$s > 0.3). Individual subjects' activation maps visualize the similarity between boundary patterns during encoding and recall (*Figure 3B*, *Figure 3—figure supplement 1*). We observed similar results in the lateral parietal DMN subregion (angular gyrus; *Figure 3—figure supplement 2*), as well as using shorter (4.5 s) time windows of boundary and non-boundary periods (*Figure 2—figure supplement 1*, *Figure 3—figure supplement 3*).

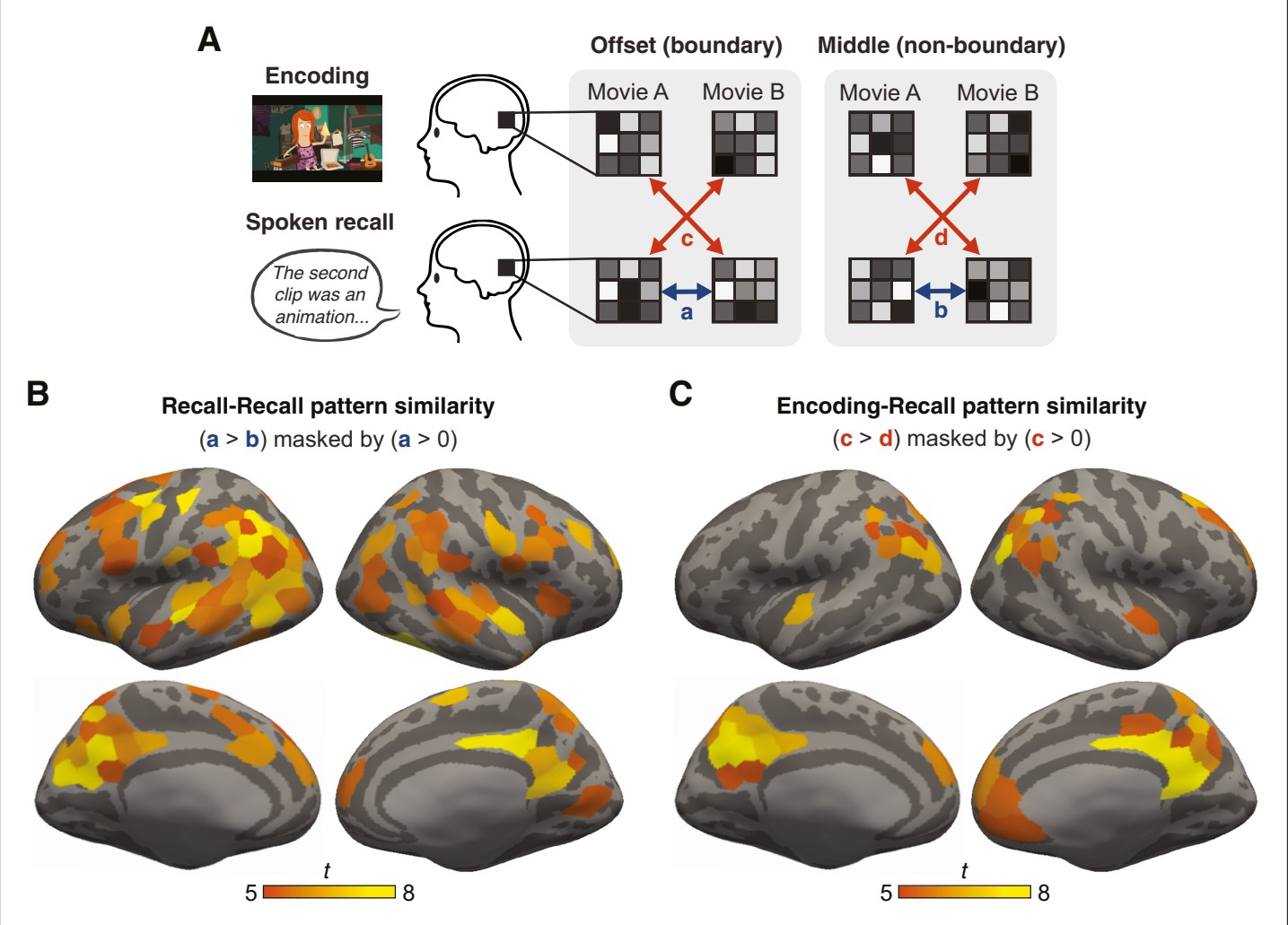

**Figure 2.** Consistent activation patterns associated with between-movie boundaries. (**A**) Schematic of the pattern similarity analysis. Boundary patterns were defined as the mean pattern averaged across 15 s following the offset of each watched or recalled movie. Non-boundary patterns were defined as the mean pattern averaged across 15 s in the middle of each watched or recalled movie. For each subject and cortical parcel (*Schaefer et al., 2018*; 200 parcels per hemisphere), we computed pairwise between-movie pattern similarity (Pearson correlation), separately for boundary patterns and non-boundary patterns measured during recall (a and b, blue arrows). We also computed between-movie and between-phase (encoding-recall) pattern similarity, again separately for boundary and non-boundary patterns (c and d, red arrows). The time windows for both boundary and non-boundary periods were shifted forward by 4.5 s to account for the hemodynamic response delay. (**B**) Whole-brain *t* statistic map of cortical parcels that showed consistent between-movie boundary patterns during recall. These parcels displayed significantly greater between-movie pattern similarity in the boundary condition compared to the non-boundary condition during recall. The map was masked by parcels that showed significantly positive between-movie pattern similarity in the boundary condition during recall. Both effects were Bonferroni corrected across parcels (p<0.05). (**C**) Whole-brain *t* statistic map of cortical parcels that showed consistent between-movie boundary patterns across encoding and recall. These parcels displayed significantly greater between-movie and between-phase pattern similarity in the boundary condition compared to the non-boundary condition. The map was masked by parcels that showed significantly positive between-movie and between-phase pattern similarity in the boundary condition. Both effects were Bonferroni corrected across parcels (p<0.05).

The online version of this article includes the following figure supplement(s) for figure 2:

**Figure supplement 1.** Consistent activation patterns during shorter (4.5 s) time windows following between-movie boundaries.

**Figure supplement 2.** Similar visual input cannot explain between-movie boundary patterns consistent across experimental phases.

Thus far, we tested boundary responses following offsets, based on prior findings that post-stimulus neural responses contribute to memory formation (*Ben-Yakov et al., 2013*; *Ben-Yakov and Dudai, 2011*; *Medvedeva et al., 2021*). However, other studies also reported neural responses specific to the onset of an episode (*Bulkin et al., 2020*; *Fox et al., 2005*; *Wen et al., 2020*). Is the generalized

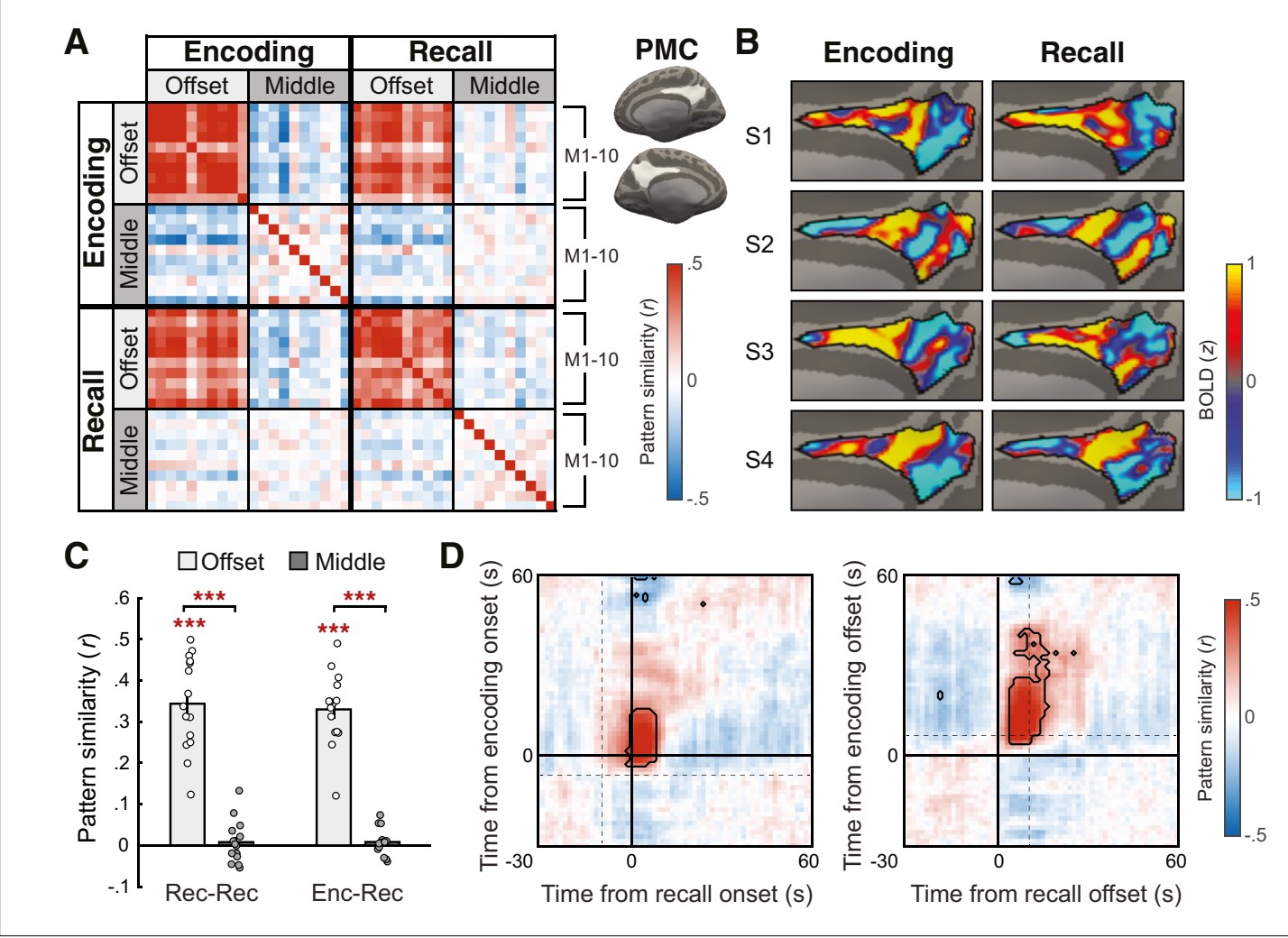

**Figure 3.** Boundary pattern in the posterior medial cortex (PMC). (**A**) PMC activation pattern similarity (Pearson correlation) between the 10 movie stimuli (M1–10), conditions (offset = boundary, middle = non-boundary), and experimental phases (encoding, recall), averaged across all subjects. The boundary pattern of a movie was defined as the mean pattern averaged across the 15 s window following the offset of the movie. The non-boundary pattern was defined as the mean pattern averaged across the 15 s window in the middle of a movie. The time windows for both boundary and non-boundary patterns were shifted forward by 4.5 s to account for the hemodynamic response delay. PMC regions of interest (ROIs) are shown as white areas on the inflated surface of a template brain. (**B**) Subject-specific mean activation patterns associated with between-movie boundaries during encoding (left) and recall (right). The boundary patterns were averaged across all movies and then z-scored across vertices within the PMC ROI mask, separately for each experimental phase. PMC (demarcated by black outlines) of four example subjects (S1–4) are shown on the medial surface of the right hemisphere of the fsaverage6 template brain. (**C**) Within-phase (recall-recall) and between-phase (encoding-recall) pattern similarity across different movies, computed separately for the boundary (offset) and non-boundary (middle) patterns in PMC. Bar graphs show the mean across subjects. Circles represent individual subjects. Error bars show SEM across subjects. ***p<.001. (**D**) Time-point-by-time-point PMC pattern similarity across the encoding phase and recall phase activation patterns around between-movie boundaries, averaged across all subjects. The time series of activation patterns were locked to either the onset (left) or the offset (right) of each movie. During encoding, the onset of a movie and the offset of the preceding movie were separated by a 6 s title scene. During recall, onsets and offsets of recalled movies were separated by, on average, a 9.3 s pause (boundaries concatenated across subjects, SD = 16.8 s). Dotted lines on the left and right panels indicate the mean offset times of the preceding movies and the mean onset times of the following movies, respectively. Note that in this figure zero corresponds to the true stimulus/behavior time, with no shifting for hemodynamic response delay. Areas outlined by black lines indicate correlations significantly different from zero after multiple comparisons correction (Bonferroni corrected p<0.05). Time–time correlations within each experimental phase can be found in *Figure 3—figure supplement 4*.

The online version of this article includes the following source data and figure supplement(s) for figure 3:

**Source data 1.** Source data for *Figure 3*.

**Figure supplement 1.** Subject-specific boundary patterns in the posterior medial cortex (PMC).

**Figure supplement 2.** Boundary pattern in the angular gyrus (ANG).

*Figure 3 continued on next page*

*Figure 3 continued*

**Figure supplement 2—source data 1.** Source data for *Figure 3—figure supplement 2*.

**Figure supplement 3.** Boundary patterns in regions of interest measured during shorter (4.5 s) time windows.

**Figure supplement 3—source data 1.** Source data for *Figure 3—figure supplement 3*.

**Figure supplement 4.** Time–time pattern similarity in the posterior medial cortex (PMC).

**Figure supplement 4—source data 1.** Source data for *Figure 3—figure supplement 4*.

boundary pattern evoked by the onset of a movie, rather than the offset? We examined this question by comparing the temporal emergence of the generalized boundary pattern following movie offsets versus onsets (*Figure 3D*); note that the offset of a movie was temporally separated from the onset of the following movie during both encoding and recall (see *Figure 1A*). Specifically, we extracted the mean time series of PMC activation patterns around between-movie boundaries, time-locked to either the onset or offset of each watched or recalled movie. We then computed between-phase (encoding-recall) pattern similarity across the individual time points of the activation pattern time series. We found that significantly positive between-phase correlations emerged well before the encoding and recall onsets (*Figure 3D*, left panel), starting from 4.5 s following the offsets of the preceding watched or recalled movie (*Figure 3D*, right panel). Thus, boundary patterns were not exclusively triggered by movie onsets; it is likely that offset responses significantly contributed to the boundary patterns.

We focused our analyses up to this point on transitions between movies because they provided clear boundaries between mental contexts during recall. However, event boundaries in naturalistic movie stimuli are often defined as transitions between scenes within a movie (*Baldassano et al., 2017*; *Chen et al., 2017*; *Zacks et al., 2010*). In prior work, it has been shown that for within-movie event boundaries neural responses scale positively with human judgments of the 'strength' of scene transitions (*Ben-Yakov and Henson, 2018*). Thus, we hypothesized that boundaries between movies (i.e., between mental contexts) would manifest as stronger versions of within-movie boundaries with qualitatively similar patterns; in other words, boundary patterns would generalize across different scales of boundaries. To test this idea, we first confirmed that there were consistent within-movie event boundary patterns in PMC during encoding; within-movie boundary patterns were more similar to each other than to non-boundary patterns (*Figure 4—figure supplement 1*). We then tested whether this within-movie boundary pattern resembled the between-movie boundary pattern by measuring the correlation between (1) the mean between-movie boundary pattern during recall and (2) the mean within-movie event boundary pattern during encoding (*Figure 4*). Surprisingly, the two were negatively correlated ($t(14) = 5.10$, p<0.001, Cohen's $d_z = 1.32$, 95% CI = [−0.34, −0.14]), in contrast to the strong positive correlation across encoding and recall between-movie boundary patterns ($t(14) = 25.02$, p<.001, Cohen's $d_z = 6.46$, 95% CI = [0.67,0.79]). The within-movie event boundary pattern was also negatively correlated with the encoding phase between-movie boundary pattern ($t(14) = 7.31$, p<0.001, Cohen's $d_z = 1.89$, 95% CI = [−0.44, −0.24]). Within-movie and between-movie boundary patterns did not resemble each other, regardless of the specific time windows used to define the boundary periods (*Figure 4—figure supplement 2*). These results suggest that the between-movie boundary pattern may reflect a cognitive state qualitatively different from the state elicited by within-movie event boundaries during movie watching.

Is the generalized between-movie boundary pattern driven by shared low-level perceptual or motoric factors rather than cognitive states? First, shared visual features at between-movie boundaries (i.e., black screen) cannot explain the transient, boundary-specific similarity between encoding and recall phases because visual input was identical across boundary and non-boundary periods during recall (i.e., a fixation dot on black background). Indeed, encoding boundary patterns were more similar to recall boundary patterns than to recall non-boundary patterns in DMN areas, suggesting a limited contribution of shared visual input to the generalized boundary pattern (*Figure 2—figure supplement 2*). Likewise, the absence of verbal responses at boundaries cannot explain the boundary pattern generalized across encoding and recall phases as no speech was generated throughout the entire encoding phase. Moreover, PMC boundary patterns showed positive between-phase pattern correlations ($t(14) = 3.94$, p=0.003, Cohen's $d_z = 1.25$, 95% CI = [0.1,0.36]) greater than those of non-boundary patterns ($t(14) = 3.22$, p=0.011, Cohen's $d_z = 1.02$, 95% CI of the difference = [0.06,0.36]) even when restricted to boundaries without pauses between recalled movies. We also ruled out the

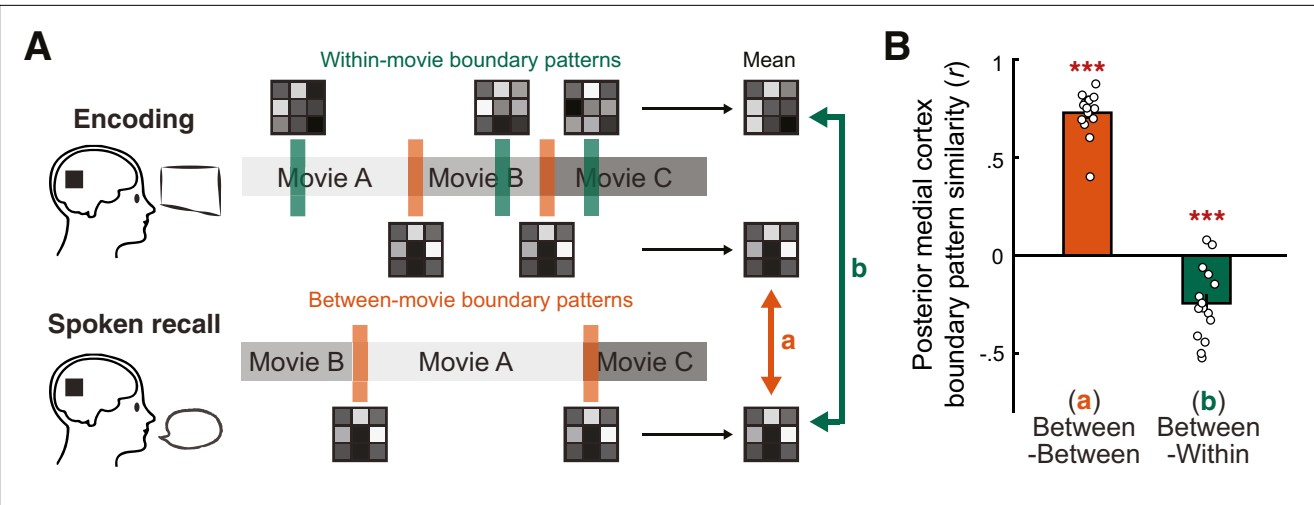

**Figure 4.** Comparing between-movie and within-movie boundary patterns in the posterior medial cortex (PMC). (**A**) Schematic of the analysis. For each subject, we created the template PMC activation pattern associated with between-movie boundaries by averaging activation patterns following the offset of each between-movie boundary (orange bars), separately for encoding and recall phases. Likewise, the template within-movie event boundary pattern was created by averaging the activation patterns following the offset of each within-movie boundary during encoding (green bars). We then measured the similarity (Pearson correlation) between the mean between-movie boundary patterns during encoding and recall (a, orange arrow). We also measured the similarity between the mean within-movie boundary pattern during encoding and the mean between-movie boundary pattern during recall (b, green arrow). For both between- and within-movie boundaries, boundary periods were 15 s long, shifted forward by 4.5 s. (**B**) Pattern similarity between template boundary patterns. The orange bar shows the mean correlation across the between-movie boundary patterns during encoding and recall. The green bar shows the mean correlation across the between-movie boundary pattern during recall and the within-movie boundary pattern during encoding. Circles represent individual subjects. Error bars show SEM across subjects. ***p<0.001 against zero.

The online version of this article includes the following source data and figure supplement(s) for figure 4:

**Source data 1.** Source data for *Figure 4*.

**Figure supplement 1.** Comparing within-movie boundary patterns and non-boundary (middle) patterns in the posterior medial cortex (PMC) during encoding.

**Figure supplement 1—source data 1.** Source data for *Figure 4—figure supplement 1*.

**Figure supplement 2.** Examining the effects of boundary period time windows on the between- and within-movie boundary pattern similarity in the posterior medial cortex (PMC).

**Figure supplement 2—source data 1.** Source data for *Figure 4—figure supplement 2*.

possibility that silence during movie title scenes and pauses at recall boundaries drove the generalized boundary pattern in PMC; the recall boundary pattern was not correlated with the pattern associated with silent periods during encoding ($t(14) = 1.93$, p=0.074, Cohen's $d_z = 0.498$, 95% CI = [–0.19,0.01]), whereas the auditory cortex showed a positive correlation between the two ($t(14) = 10.31$, p<0.001, Cohen's $d_z = 2.66$, 95% CI = [0.3,0.45]) (*Figure 5*). Likewise, the movies' audio amplitudes modulated the time course of similarity between the recall boundary pattern and the encoding data in the auditory cortex, but not in PMC (*Figure 5—figure supplements 1 and 2*).

## Discussion

This study investigated brain responses to internally generated boundaries between mental contexts during continuous and unguided memory recall of naturalistic narratives. We found that internally driven mental context boundaries evoke generalized neural activation patterns in core posterior-medial areas of the DMN (*Ritchey and Cooper, 2020*). These cortical patterns were similar to those observed at major boundaries between externally driven contexts (different audiovisual movies), suggesting that they reflect a general cognitive state associated with mental context transitions. However, these between-context patterns were distinct from within-context event boundary detection signals.

The highly similar neural activation patterns for internally- and externally driven boundaries observed in this study demonstrate event segmentation without changes in external input. This finding diverges

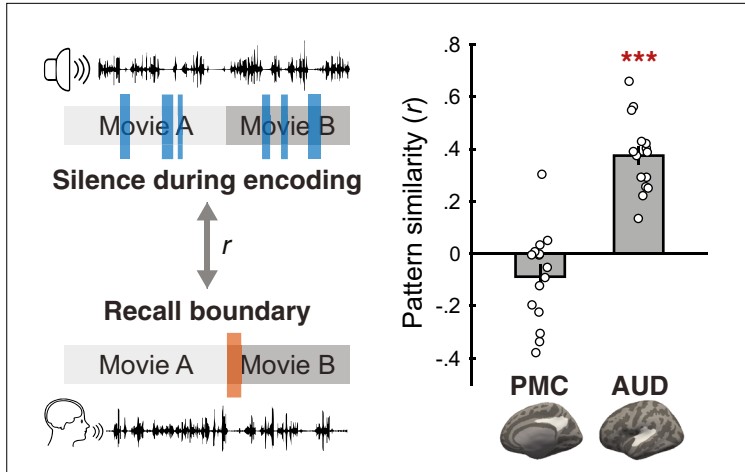

**Figure 5.** Examining the effects of silence on the generalized boundary pattern. For each subject, we computed a Pearson correlation between the mean activation pattern of the moments of silence during encoding (blue bars) and the mean activation pattern of between-movie boundaries during recall (orange bar) in the posterior medial cortex (PMC) and the auditory cortex (AUD). The moments of silence near between-movie boundaries (i.e., within the first 45 s of each movie) during encoding were excluded from the analysis. PMC and AUD regions of interest are shown as white areas on the inflated surface of template brains. Gray bars on the right panel indicate the mean pattern similarity across subjects. Circles represent individual subjects. Error bars show SEM across subjects. ***p<0.001 against zero.

The online version of this article includes the following source data and figure supplement(s) for figure 5:

**Source data 1.** Source data for *Figure 5*.

**Figure supplement 1.** Time series of audio amplitudes during encoding and the similarity to the recall boundary pattern.

**Figure supplement 1—source data 1.** Source data for *Figure 5—figure supplement 1*.

**Figure supplement 2.** Relationship between audio amplitudes during encoding and the similarity to the recall boundary pattern.

**Figure supplement 2—source data 1.** Source data for *Figure 5—figure supplement 2*.

from the currently dominant empirical and theoretical perspectives on event segmentation; in most studies, event boundaries are defined or manipulated by changes in perceptual or spatiotemporal features (e.g., *Chen et al., 2017*; *DuBrow and Davachi, 2013*; *Radvansky and Copeland, 2006*), and boundary detection is posited to occur when those changes mismatch our expectations of the current situation (*Zacks et al., 2007*; *Zacks et al., 2011*). This prediction error framework successfully explains various phenomena related to event perception and memory organization (see *Zacks, 2020* for a review); however, evidence has also shown that predicted changes in external features can create boundaries and have similar behavioral effects (*Pettijohn and Radvansky, 2016*; *Schapiro et al., 2013*). To resolve the discrepancy, an alternative theoretical framework has recently proposed that boundaries are perceived when the probability distribution of inferred current situations, rather than observed external features per se, changes from the previous time point (*Shin and DuBrow, 2021*). According to this account, event segmentation can occur when there is no perceptual change or when transitions are already predicted, which may explain the boundary-related neural responses at self-generated transitions between memories during recall in our study.

The boundary pattern that generalized across internally and externally driven boundaries was most strongly observed in the DMN, in line with earlier findings implicating the DMN in mental context transitions (*Baldassano et al., 2017*; *Crittenden et al., 2015*; *Smith et al., 2018*). Prior studies have shown that the DMN responds to external context transitions, including experimental task switching (*Crittenden et al., 2015*; *Smith et al., 2018*) as well as event boundaries in movie clips (*Reagh et al., 2020*; *Speer et al., 2007*). Considering these findings and the widely known involvement of the DMN in internally oriented cognition (e.g., *Addis et al., 2007*; *Andrews-Hanna et al., 2010*; *Christoff et al., 2009*) together, it has been suggested that the DMN integrates both internal and

external information to represent and maintain an abstract mental model of the current situation or state (*Stawarczyk et al., 2021*; *Yeshurun et al., 2021*); located furthest away from sensorimotor areas (*Smallwood et al., 2021*), the DMN integrates information across different modalities (*Bonnici et al., 2016*; *Ramanan et al., 2018*) and over long timescales (*Chang et al., 2021*; *Hasson et al., 2015*). Supporting this idea, neural activation patterns in subregions of the DMN, especially PMC, tend to persist for extended periods of time during naturalistic movie watching, and transitions between these persistent neural states coincide with perceived event boundaries (*Baldassano et al., 2017*; *Geerligs et al., 2021*). Our study extends this finding by identifying a transient, boundary-induced phenomenon, which is a unique and independent state represented in the DMN. That is, at major event boundaries, a temporary boundary state may exist in between the neural patterns representing the two events, rather than one event pattern switching directly to the next.

Although the boundary-related PMC activation patterns were consistent across internally and externally driven boundaries, they did not generalize across within- and between-movie boundaries. Relatedly, a recent human neurophysiological study (*Zheng et al., 2022*) reported that medial temporal cortex neurons distinguished within- and between-movie boundaries while subjects were watching short video clips; some neurons responded only to between-movie boundaries, whereas a separate group of neurons responded to both types of boundaries. These findings may be in line with the view that event boundaries have a hierarchical structure, with different brain areas along the information pathway reflecting different levels of boundaries, from fine-grained sensory transitions to coarse-grained situational transitions (*Baldassano et al., 2017*; *Chang et al., 2021*; *Geerligs et al., 2021*). However, it is still puzzling that within- and between-movie boundaries in our study produced qualitatively distinct neural patterns within a highest-order area (PMC), even though both categories consisted of prominent boundaries between situations spanning tens of seconds to several minutes. What are the crucial differences between the two levels of boundaries? One important factor might be the presence or absence of inter-event connections. Even the most salient within-movie boundaries still demand some integration of information across events as the events are semantically or causally related, and ultimately constitute a single coherent narrative (*Lee and Chen, 2021*; *Song et al., 2021b*). In contrast, an entire cluster of related events, or the narrative as a whole, might be completely 'flushed' at between-movie boundaries; this difference could induce distinct cognitive states at the two levels of boundaries, giving rise to different PMC patterns.

What is the cognitive state that is generalized across internal- and external boundaries between completely different contexts, but distinct from the state evoked by boundaries within the same context? We speculate that the between-movie boundary state may be a temporary 'relay' state that occurs when no one mental model wins the competition to receive full attentional focus following the flushing of the prior mental context. Namely, when one major mental context switches to another, the brain may pass through a transient off-focus (*Mittner et al., 2016*) or mind-blanking (*Mortaheb et al., 2021*; *Ward and Wegner, 2013*) state that is distinct from both processing external stimuli (e.g., movie watching) and engaging in internal thoughts (e.g., memory recall). This account may also explain the difference between within- vs. between-movie boundary patterns: in terms of attentional fluctuation (*Jayakumar et al., 2022*; *Song et al., 2021a*), external attention is enhanced at within-movie event boundaries (*Pradhan and Kumar, 2021*; *Zacks et al., 2007*), whereas the relay state is associated with lapses in attention (*deBettencourt et al., 2018*; *Esterman et al., 2014*). An alternative, but not mutually exclusive, possibility is that the boundary state involves the recruitment of cognitive control to resolve the competition between mental contexts. This idea is based on the observation that the areas showing relatively higher activation at between-movie boundaries overlap with the frontoparietal control network (FPCN; *Vincent et al., 2008*) both during encoding and recall (*Figure 1B*, *Figure 1—video 2*). As the FPCN is interdigitated with the DMN and other nearby areas within individual subjects (*Braga and Buckner, 2017*), relative activation of the FPCN may create the stereotyped boundary pattern in higher associative cortices. It is also noteworthy that both of these candidate cognitive states are triggered not by the onset but by the offset of a mental context; the onset would rather signal the resolution of competition between mental contexts, hence the end of those states. This dovetails with our results showing that the generalized boundary pattern appears well before movie onsets, suggesting a major contribution of offset responses.

In conclusion, we found that internally driven boundaries between memories produce a stereotyped activation pattern in the DMN, potentially reflecting a unique cognitive state associated with

the flushing and updating of mental contexts. By demonstrating stimulus-independent event segmentation during continuous and naturalistic recall, our study bridges the gap between the fields of event segmentation and spontaneous internal thoughts (also see *Tseng and Poppenk, 2020*). Without any task demands or external constraints, the mind constantly shifts between different internal contexts (*Raffaelli et al., 2021*; *Sripada and Taxali, 2020*). What are the characteristics of neural responses to different types of spontaneous mental context boundaries (e.g., between two different memories, between external attention and future thinking)? Is the boundary pattern observed in this study further generalizable to mental context transitions even more stark than between-movie transitions in our experiment? Are there specific neural signatures that predict subsequent thought transitions? Future work will explore answers to these questions by employing neuroimaging methods with behavioral paradigms that explicitly and continuously track the unconstrained flow of thoughts in naturalistic settings.

## Materials and methods

Here, we provide a selective overview of procedures and analysis methods. More detailed descriptions of participants, stimuli, experimental procedures, fMRI data acquisition, and preprocessing can be found in *Lee and Chen, 2021*.

### Participants

Twenty-one subjects (12 females) between the ages of 20 and 33 participated in the study. Informed consent was obtained in accordance with procedures approved by the Princeton University Institutional Review Board (protocol #5516). Six subjects were excluded from analyses due to excessive motion.

### Stimuli

Ten audiovisual movies (range 2.15–7.75 min) were used in the experiment. The movies varied in format (animation, live-action) and content. Each movie clip was prepended with a title scene where the movie title in white letters faded in and out at the center of the black screen. The movie title was shown approximately for 3 s of the 6-s-long title scene. At the beginning of each scanning run, a 39-s-long audiovisual introductory cartoon was played before the movie stimuli. The introductory cartoon was excluded from analyses.

### Experimental procedures

The experiment consisted of two phases, encoding and free spoken recall (*Figure 1A*), both performed inside the MRI scanner. In the encoding phase, subjects watched a series of 10 short movies. Subjects were instructed to pay attention to the movies, and no behavioral responses were required. There were two scanning runs, and subjects watched five movies in each run. Stimulus presentation began 3 s after the first volume of each run. In the free spoken recall phase, subjects were instructed to verbally recount what they remembered from the movies, regardless of the order of presentation. Subjects were encouraged to describe their memory in their own words in as much detail as possible. A white dot was presented in the center of the black screen during the free spoken recall phase, though subjects were not required to fixate. The recall phase consisted of two scanning runs in 4 of the 15 subjects included in the analysis. The other subjects had a single scanning run. Subjects' recall speech was audio-recorded using an MR-compatible noise-canceling microphone and then manually transcribed. The recall transcripts were also timestamped to identify the onset and offset of the description of each movie (there were no intrusions across movies during recall).

### fMRI acquisition and preprocessing

Imaging data were collected on a 3 T Siemens Prisma scanner at Princeton Neuroscience Institute. Functional images were acquired using a T2*- weighted multiband accelerated echo-planar imaging sequence (TR = 1.5 s; TE = 39 ms; flip angle = 50°; acceleration factor = 4; 60 slices; $2 \times 2 \times 2$ mm$^3$). Whole-brain anatomical images and fieldmap images were also acquired. Functional images were motion-corrected and unwarped using FSL, and then coregistered to the anatomical image, resampled to the fsaverage6 cortical surface, and smoothed (FWHM 4 mm) using FreeSurfer Functional

Analysis Stream. The smoothed data were also high-pass filtered (cutoff = 140 s) and z-scored within each scanning run. The first five volumes of encoding scanning runs and the first three volumes of free spoken recall scanning runs were excluded from analyses.

## Cortical parcellation and region of interest (ROI) definition

For whole-brain pattern similarity analysis, we used an atlas (*Schaefer et al., 2018*) that divided the cortical surface into 400 parcels (200 parcels per hemisphere) based on functional connectivity patterns (17 networks version). For ROI analyses, we defined the bilateral PMC by combining the parcels corresponding to the precuneus and posterior cingulate cortex within Default Network A as in our prior study (*Lee and Chen, 2021*). The precuneus and posterior cingulate cortex together spanned the area that showed the strongest content-and task-general boundary patterns in the whole-brain analysis (*Figure 3C*). The bilateral angular gyrus ROI consisted of the parcels corresponding to the inferior parietal cortex within Default Network A, B, and C. The bilateral auditory cortex ROI was defined by combining the parcels corresponding to the primary and secondary auditory cortices within Somatomotor Network B.

## Univariate activation analysis

We performed whole-brain univariate activation analysis to identify brain areas that show activation changes at between-movie boundaries compared to non-boundary periods during recall (*Figure 1B*). The boundary periods were the first 15 s following the offset of each recalled movie, and the non-boundary periods were the 15 s in the middle of each recalled movie. Both boundary and non-boundary period time windows were shifted forward by 4.5 s to account for the hemodynamic response delay. We used a relatively long 15 s duration for the boundary and non-boundary periods to capture most of the boundary-related signals during recall, based on exploratory analyses that examined the time courses of univariate boundary responses (*Figure 1—figure supplement 1*) and boundary-triggered activation patterns (*Figure 3—figure supplement 4D*). For each vertex in each subject's brain, we computed the mean boundary activation by first averaging preprocessed BOLD signals across time points within each boundary period, and then across all recalled movies. Likewise, we computed the mean non-boundary activation for each subject and vertex by first averaging preprocessed BOLD signals across time points within each non-boundary period, and then across all recalled movies. We then computed the difference between the boundary and non-boundary activation for each subject. Finally, we performed a group-level one-sample $t$-test against zero (two-tailed). The Benjamini–Hochberg procedure (false discovery rate $q < 0.05$) was applied to correct for multiple comparisons across vertices on the resulting whole-brain statistical parametric map.

## Pattern similarity analysis

We performed whole-brain pattern similarity analysis (*Figure 2A*) to identify brain areas that showed content-and task-general neural activation patterns associated with between-movie boundaries. For each cortical parcel of each subject's brain, we extracted boundary and non-boundary activation patterns for each movie, separately for the encoding phase and the recall phase. Boundary patterns were generated by averaging the spatial patterns of activation within the boundary period (the first 15 s following the offset) of each watched or recalled movie. Non-boundary patterns were generated by averaging spatial patterns within the non-boundary period (the middle 15 s) of each watched or recalled movie. Again, both boundary and non-boundary time windows were shifted forward by 4.5 s to account for the hemodynamic response delay. We then computed Pearson correlation coefficients between the patterns within and across different movies, conditions (boundary, non-boundary), and experimental phases.

Using the resulting correlation matrix (see *Figure 3A* for an example) for each parcel, we first identified brain areas that showed boundary patterns that were consistent *across recalled movies* (*Figure 2B*). For each subject's recall phase, we computed the mean of all pairwise between-movie correlations, separately for the boundary patterns and the non-boundary patterns. We then performed a group-level two-tailed one-sample $t$-test against zero on the mean boundary pattern correlations to test whether the boundary pattern similarity was overall positive. We also performed a group-level two-tailed paired-samples $t$-test between the mean boundary vs. non-boundary pattern correlations to test whether the boundary pattern similarity was greater than the non-boundary pattern similarity.

Each of the resulting whole-brain statistical parametric maps was corrected for multiple comparisons across parcels using the Bonferroni method. Finally, we identified parcels that showed significant effects in both tests after the correction by masking the areas that showed higher pattern similarity for the boundary than non-boundary conditions with the areas that showed overall positive similarity between boundary patterns (*Figure 2B*). Thus, the identified parcels showed spatially similar activation patterns across different movies at recall boundaries, and the patterns were specifically associated with boundary periods only. Likewise, we identified brain areas that showed boundary patterns that were consistent *across the encoding and recall phases* as well as across movies (*Figure 2C*). This was achieved by repeating the identical analysis procedures using the boundary and non-boundary pattern correlations computed across the encoding and recall phases, instead of using the correlations computed within the recall phase.

We also performed the same pattern similarity analysis in the PMC (*Figure 3*) and angular gyrus (*Figure 3—figure supplement 2*) ROIs, as done for an individual cortical parcel in the whole-brain analysis. In addition, we repeated the same analyses using shorter (4.5 s) boundary and non-boundary period time windows and obtained similar results (*Figure 2—figure supplement 1*, *Figure 3—figure supplement 3*).

## Comparing the onset- and offset-locked boundary patterns

To test whether the consistent activation patterns associated with between-movie boundaries were evoked by the onset or offset of a movie, we examined TR-by-TR pattern correlations across time points around the boundaries. The time points were locked to either the onset or the offset of (1) each video clip (excluding the title scene) or (2) recall of each movie. For each subject and ROI, we extracted the time series of activation patterns from 30 s before to 60 s after the onset/offset of each watched or recalled movie. We averaged the time series across movies to create a single time series of boundary-related activation patterns per experimental phase. We then computed Pearson correlation coefficients across different time points in the time series of mean activation patterns within each experimental phase (i.e., encoding-encoding and recall-recall correlation; *Figure 3—figure supplement 4*) or between phases (i.e., encoding-recall correlation; *Figure 3D*, *Figure 3—figure supplement 2D*). Finally, we performed two-tailed one-sample *t*-tests against zero on each cell of the time–time correlation matrices from all subjects to identify the time points at which significantly positive or negative pattern correlations appeared. Bonferroni correction was applied to correct for multiple comparisons across all cells in the time–time correlation matrix.

## Comparing the between-movie and within-movie boundary patterns

To test whether the recall activation patterns evoked by between-movie boundaries were similar to encoding activation patterns evoked by event boundaries within a movie, we identified the strongest event boundaries within each movie. We utilized the fine-grained event boundaries defined in our previous study (*Lee and Chen, 2021*), which divided the 10 movie stimuli into 202 events excluding title scenes (mean duration = 13.5 s, range 2–42 s). We had four independent coders watch the movie stimuli and then choose which of the fine-grained event boundaries were the most important. The coders were instructed to select the boundaries such that the 10 movies were divided into 60 ± 10 events excluding title scenes. Of these, 25 event boundaries were identified as important by all four coders, which resulted in 27 'coarse' events in total (ranging between 1 and 5 events per movie; mean duration = 100.9 s, range 21–417 s). To mitigate the possibility of carryover effects from the between-movie boundaries, within-movie event boundaries that occurred within the first 45 s of each movie clip were excluded from the analysis, leaving 15 within-movie event boundaries in total.

We first examined whether there were consistent activation patterns following the within-movie event boundaries distinct from non-boundary patterns (*Figure 4—figure supplement 1*). For each subject, we generated the mean PMC activation pattern for each within-movie boundary by averaging patterns from 4.5 to 19.5 s following the within-movie boundary during encoding. We then computed pairwise between-movie Pearson correlations across the within-movie boundary patterns and averaged the correlations. A two-tailed one-sample *t*-test against zero was performed to test whether the similarity between the within-movie boundary patterns was overall positive. We also computed pairwise between-movie correlations across the within-movie boundary patterns and non-boundary patterns during encoding. The non-boundary pattern for each movie was generated by averaging activation

patterns within the middle 15 s of the movie (time window shifted forward by 4.5 s). A two-tailed paired-samples *t*-test was performed to test whether the similarity between within-movie boundary patterns was greater than the similarity between within-movie boundary patterns and non-boundary patterns. Two of the non-boundary periods partially overlapped with the within-movie boundary periods by 13.5 and 4.5 s, respectively, and were excluded when correlating within-movie boundary patterns and non-boundary patterns. Note that the two non-boundary periods were included in other analyses in this study comparing between-movie boundary patterns and non-boundary patterns. However, excluding or including the two non-boundary periods did not significantly change any of the mean pairwise between-movie correlations across (1) encoding non-boundary patterns, (2) encoding non-boundary and between-movie boundary patterns, (3) encoding non-boundary and recall non-boundary patterns, and (4) encoding non-boundary and recall between-movie boundary patterns in PMC (two-tailed paired-samples *t*-tests, all $t(14)$s < 1.45, all ps>0.17).

We next compared the template activation pattern at the within-movie event boundaries to the pattern at between-movie boundaries (*Figure 4*). For each subject, we generated the mean within-movie event boundary pattern of PMC by averaging activation patterns from 4.5 to 19.5 s following each of the 15 event boundaries during encoding. The patterns were first averaged across all time points within each boundary period time window and then across different boundaries. Likewise, the mean between-movie boundary pattern was generated by averaging all activation patterns from 4.5 to 19.5 s following the offset of each movie during encoding or recall. We then computed a Pearson correlation coefficient across the mean within-movie event boundary pattern and the mean encoding or recall between-movie boundary pattern. For comparison, we computed a correlation across the encoding and recall mean between-movie boundary patterns. A two-tailed one-sample *t*-test against zero was performed to test whether the group-level similarity between the two patterns was positive. We also repeated the same pattern similarity analysis using shorter (4.5 s) time windows for the boundary periods, from 4.5 to 9 s following the within- or between-movie boundaries (*Figure 4—figure supplement 2A*).

To explore the temporal unfolding of the similarity between the within- and between-movie boundary patterns, we additionally examined the between-phase TR-by-TR pattern similarity across individual time points around the boundaries (*Figure 4—figure supplement 2B*). For each subject, we extracted the PMC activation pattern time series from 30 s before to 60 s after (1) each within-movie event boundary during encoding and (2) the offset of each movie during recall. The time series were averaged across boundaries within each experimental phase. We then computed Pearson correlation coefficients across different time points in the activation pattern time series between the encoding and recall phases. Finally, we performed two-tailed one-sample *t*-tests against zero on each cell of the time–time correlation matrices from all subjects to identify the time points at which significantly positive or negative pattern correlations appeared. Bonferroni correction was applied to correct for multiple comparisons across cells.

## Testing the effect of visual features

Between-movie boundary periods during encoding and those during recall shared low-level visual features (i.e., mostly blank black screen). To test whether the similar visual features produced similar activation patterns at between-movie boundaries across phases, we performed a whole-brain pattern similarity analysis (*Figure 2—figure supplement 2*). For each subject and cortical parcel, we computed the mean boundary and non-boundary activation patterns for each movie, separately for encoding and recall. The boundary periods were defined as the first 15 s following the offset of each watched or recalled movie. The non-boundary periods were defined as the middle 15 s of each movie. Both boundary and non-boundary time windows were shifted forward by 4.5 s. We then computed Pearson correlations between encoding boundary patterns and recall boundary patterns across different movies, and averaged all the correlations. Likewise, we computed the average correlation between boundary patterns during encoding and non-boundary patterns during recall across different movies. A group-level two-tailed paired-samples *t*-test was performed to test whether the similarity between encoding and recall boundary patterns was greater than the similarity between encoding boundary patterns and recall non-boundary patterns, even though boundary and non-boundary patterns were visually identical during recall. The resulting whole-brain map was corrected for multiple comparisons across parcels using the Bonferroni method.

## Testing the effect of audio amplitudes

Brief periods of silence were present at transitions between movies during both encoding and recall. During encoding, the 6 s title period between movies was silent. During recall, subjects often paused speaking for several seconds between recall of different movies. We tested whether the between-movie boundary patterns were associated with the absence of sound in general, as opposed to between-movie transitions specifically.

We first compared the activation pattern associated with any silent periods within movies during encoding and the activation pattern evoked by between-movie boundaries during recall (*Figure 5*). To identify all periods of silence within the movies, we extracted the audio amplitudes of the movie clips (*Figure 5—figure supplement 1A*) by applying a Hilbert transform to the single-channel audio signals (44.1 kHz). The audio amplitudes were downsampled to match the temporal resolution of fMRI data (TR = 1.5 s), convolved with a double-gamma hemodynamic response function, and *z*-scored across time points. The periods of silence were defined as the within-movie time points (again excluding the first 45 s of each movie) whose audio amplitudes were equal to or lower than the mean amplitude of the time points corresponding to the silent between-movie title periods. For each subject and ROI, we averaged the activation patterns across all time points within these within-movie silent time periods to produce the mean activation pattern associated with the absence of sound. The mean pattern was then correlated with the template between-movie boundary pattern produced by averaging 4.5–19.5 s following the offset of each movie during recall. A two-tailed one-sample *t*-test was performed to compare the group-level correlation coefficients against zero.

We additionally tested whether the time course of audio amplitude was correlated with the time course of pattern similarity (Pearson correlation) between the recall phase between-movie boundary pattern and each time point of the encoding phase data (*Figure 5—figure supplement 1B and C*, *Figure 5—figure supplement 2*). The time courses were generated for all time points within each movie, excluding the first 45 s of each movie. We first computed each subject's Pearson correlation coefficient between the two types of time courses. We then performed a group-level one-sample *t*-test against zero (two-tailed).

## Citation diversity statement

Recent work in several fields of science has identified a bias in citation practices such that papers from women and other minority scholars are under-cited relative to the number of such papers in the field (*Caplar et al., 2017*; *Dion et al., 2018*; *Dworkin et al., 2020*; *Maliniak et al., 2013*; *Mitchell et al., 2013*). Here, we sought to proactively consider choosing references that reflect the diversity of the field in thought, form of contribution, gender, race, ethnicity, and other factors. First, we obtained the predicted gender of the first and last author of each reference by using databases that store the probability of a first name being carried by a woman (*Dworkin et al., 2020*; *Zhou et al., 2020*). By this measure (and excluding self-citations to the first and last authors of this article), our references contain 14.52% women (first)/women (last), 16.13% men/women, 31.26% women/men, and 38.09% men/men. This method is limited in that (a) names, pronouns, and social media profiles used to construct the databases may not, in every case, be indicative of gender identity and (b) it cannot account for intersex, non-binary, or transgender people. Second, we obtained predicted racial/ethnic category of the first and last author of each reference by databases that store the probability of a first and last name being carried by an author of color (*Ambekar et al., 2009*; *Sood and Laohaprapanon, 2018*). By this measure (and excluding self-citations), our references contain 7.21% author of color (first)/author of color (last), 14.37% white author/author of color, 23.51% author of color/white author, and 54.91% white author/white author. This method is limited in that (a) names, Census entries, and Wikipedia profiles used to make the predictions may not be indicative of racial/ethnic identity, and (b) it cannot account for indigenous and mixed-race authors, or those who may face differential biases due to the ambiguous racialization or ethnicization of their names. We look forward to future work that could help us to better understand how to support equitable practices in science.

## Acknowledgements

We thank Sarah DuBrow, Christopher J Honey, and Megan T deBettencourt for comments on earlier versions of the manuscript. We also thank Yiyuan Zhang for assisting with within-movie event boundary

identification. This work was supported by the Sloan Research Fellowship (JC) and Google Faculty Research Award (JC).

## Additional information

### Funding

| Funder | Grant reference number | Author |
|---|---|---|
| Alfred P. Sloan Foundation | Sloan Research Fellowship | Janice Chen |
| Google | Google Faculty Research Award | Janice Chen |

The funders had no role in study design, data collection and interpretation, or the decision to submit the work for publication.

### Author contributions

Hongmi Lee, Conceptualization, Data curation, Formal analysis, Methodology, Project administration, Software, Visualization, Writing - original draft, Writing - review and editing; Janice Chen, Data curation, Funding acquisition, Investigation, Project administration, Resources, Supervision, Writing - review and editing

### Author ORCIDs

Hongmi Lee  http://orcid.org/0000-0001-8023-0727

### Ethics

Informed consent was obtained in accordance with procedures approved by the Princeton University Institutional Review Board (Protocol #5516).

### Decision letter and Author response

Decision letter https://doi.org/10.7554/eLife.73693.sa1
Author response https://doi.org/10.7554/eLife.73693.sa2

## Additional files

### Supplementary files

• Transparent reporting form

### Data availability

The raw neuroimaging and behavioral data analyzed in the current study are publicly available via OpenNeuro (https://doi.org/10.18112/openneuro.ds004042.v1.0.0). Source data files have been provided for Figure 1-figure supplement 3, Figure 3, Figure 3-figure supplements 2, 3, & 4, Figure 4, Figure 4-figure supplements 1 & 2, Figure 5, and Figure 5-figure supplements 1 & 2.

The following previously published dataset was used:

| Author(s) | Year | Dataset title | Dataset URL | Database and Identifier |
|---|---|---|---|---|
| Lee H, Chen J, Hasson U | 2022 | FilmFestival | https://doi.org/10.18112/openneuro.ds004042.v1.0.0 | OpenNeuro, 10.18112/openneuro.ds004042.v1.0.0 |

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
