## [Editor Report]

This paper provides convincing evidence that internally generated event boundaries occurring at abrupt shifts in mental state evoke similar neural responses as those triggered by a change in sensory input. Given that much past work has linked the detection of event boundaries to the discrepancy between prediction and input, these new findings are significant and anticipated to spur much future research on event boundaries in the absence of external change. This innovative and methodologically rigorous study will be of interest to cognitive neuroscientists working on topics broadly related to memory, event segmentation, and mental context.

---

## [Decision Letter]

**Decision letter after peer review:**

Thank you for submitting your article "A generalized cortical activity pattern at internally-generated mental context boundaries during unguided narrative recall" for consideration by *eLife*. Your article has been reviewed by 3 peer reviewers, including Margaret L Schlichting as the Reviewing Editor and Reviewer #1, and the evaluation has been overseen by Chris Baker as the Senior Editor. The following individuals involved in review of your submission have agreed to reveal their identity: Zachariah M. Reagh (Reviewer #2); Linda Geerligs (Reviewer #3).

Essential revisions:

1. Experiential similarity with the blank screen: How might the similarity of experience (black screen, no auditory input) across boundary periods be driving the analysis? Please consider this in a revision. Specifically, the authors should: (a) revise (perhaps correct, or at least clarify) statements in the text that seem to suggest consistent input across boundary and non-boundary periods (see comments from Reviewer #1 and #3 for more on this point). Moreover, (b) please perform additional analyses to control for this confound. Reviewers had specific suggestions for how the authors might perform these controls, but the authors are also welcome to carry out these controls in a different way, as they see fit.

2. Selection of temporal epochs for both boundary and non boundary time periods: All Reviewers were concerned about the long boundary period (15s) – they had questions including why it was defined this way, whether that decision matters, and how it might cloud interpretations of comparisons across these different time period (since, for example, we might expect boundary signatures to be more transient for within-movie boundaries than between-movie ones). The Reviewers have conferred on this issue and suggest the authors use a shorter temporal window to address this comment (rather than e.g., provide justification in the text) as suggested by Reviewer #3, given the expected transience of the univariate response.

3. Concerns about within-run pattern similarity analysis: It would be important to know whether the results hold when limiting to only cross-run comparisons (see more from Reviewer #1). Moreover, please clarify in the revision how many runs there were for the recall task.

4. Providing more background for the focus on PMC: What was the reason for the focus on PMC? Reviewer #1 was not sure whether the results were specific to the PMC, and/or whether and why this was an a priori ROI. Are the operations thought to be unique to the PMC, or shared with other DMN regions? Also, Reviewer #2 was not sure why PCC and precuneus were combined to make PMC rather than considered separately. Please provide more on the logic behind these choices.

5. Univariate responses to boundaries: The description of the results in Figure 1B do not seem to match with the results show in the figure. There seem to be widespread transient changes and a larger number of vertices that show a decline in BOLD activity after a boundary. In addition, there seems to be a strong positive BOLD response throughout the default mode network. Please clarify.

6. Paper format: Reviewers felt the paper would be better suited for a longer format article, with separate Results and Discussion sections. That would allow for a more extensive discussion (see following points) of what the results might mean for our current understanding of how event segmentation works.

7. Discussion: Reviewers felt the Discussion was too short, and felt the authors' interesting and thought-providing findings warranted a more extensive unpacking. Reviewers would like to see a more fleshed-out exploration of what this general cognitive state that may exist between events might look like, and what it might be doing.

8. Also to consider in the expanded the Discussion: It is very unclear at this point whether the observed pattern of neural responses is related to the specific setup of the study or whether subtle shifts in context versus more intense transitions involve fundamentally different mechanism. I understand that this question probably cannot be answered with the current dataset, but I would appreciate some additional discussion related to this question, as well suggestions for how this could be clarified in future studies.

9. Figure 4 results presentation: Regarding the correlations between the average between-movie boundary pattern and (1) between-movie boundaries, (2) moments of silence, (3) within-movie boundaries (i.e., the data displayed in Figure 4), we had to re-read this figure caption and Results section a few times to feel that we really grasped what comparisons were being made, and how these correlations were being conducted. We think this could stand to be tweaked to be clearer, with perhaps a conceptual diagram like the one in Figure 2A included to aid in understanding.

10. Clarifying no-sound condition: Following from the above point, we were initially not totally clear on the relevance or importance of the 'no sound' condition. Upon reading further into the Results, it made sense, but seeing as this figure is referenced well before this condition is expanded upon, we think this could be motivated a little earlier to mitigate unnecessary confusion. As of now, this set of analyses seems to be motivated after presentation of the data rather than before, and simply shifting this around could aid in understanding.

11. Figure 3 results presentation: I was not sure I fully understood the offset vs. onset yoking analysis-both how it was performed, and how the conclusions followed from the results. First, I was a bit confused about how the difference in delay duration between movies at encoding (6s) versus at recall (9.3s on average, but variable; see also comment #5) would play into this and whether those are meaningful time points to display on the Figure 3D charts that might help the reader interpret those findings. Second, the authors state that this analysis shows the boundary patterns were driven by offset (more than onset) responses, but I was not sure what aspect of the results led to that conclusion. Can the authors say more about the evidence supporting this conclusion? It looks to me like there are strong correlations that emerge both after offset and onset (i.e., just above and to the right of the origin there are numerous time points with positive [red] correlations). Perhaps it is because the red positive correlations start earlier, prior to the recall itself, when yoked to the onset, but I am not sure why this means it is related to offset and not some preparation for the onset of recall (see also comment #5). Also, is it interesting or meaningful that the patterns seem more compressed at recall than at encoding (i.e., the outlined red areas are skinnier than they are tall)? Please clarify.

12. Reasoning behind masking decisions: I am not sure the reason for masking Figure 2B and C with the a>0 and c>0 maps. First of all, it seems as though in RSA the actual correlation being positive or negative is not terribly meaningful and can depend on preprocessing decisions, etc. In addition to that potential issue though I'm also just generally interested in more understanding the logic behind this decision. Can the authors explain that and include it in the main paper? Were there any regions that showed for example a

13. Clarity suggestions: The Reviewers also had several other suggestions that might increase the clarity of results presentation and/or depictions. The authors should please consider these suggestions and implement if they see fit.

*Reviewer #1 (Recommendations for the authors):*

1. I had a couple more specific comments and ideas/suggestions:

a) Regarding p. 6, line 218-219: "boundary and non-boundary periods were identical in terms of visual input during recall and speech generation during movie watching": Please clarify this phrasing. I thought boundary periods would be a blank screen, and non-boundary would be a movie playing during encoding and have verbal recall happening during recall. How is it true that they are identical in terms of visual input?

b) For overcoming the concern about visual input, I am wondering if perhaps the authors might be able to leverage the title screen before the very first movie in a run as a control, since this is not really a "boundary"? Alternatively, perhaps duration of blank screen viewing could be useful as a control since if what this is picking up on is blank-screne-ness, maybe it would increase with more time spent looking at the blank screen(?).

2. I have a number of clarification questions-these are mainly methodological choices that I simply didn't understand, and felt that further explanation or justification for these decisions should be included in the paper:

a) What was the reason for the focus on PMC? I was not sure whether the results were specific to the PMC, and/or whether and why this was an a priori ROI. Are the operations thought to be unique to the PMC, or shared with other DMN regions? More logic behind this choice would be very helpful.

b) The run structure was a bit unclear to me. Sorry if I missed this, but I saw there were two encoding scans but could not work out how many recall scans there were.

c) For the non-boundary comparisons, this was described as being the middle 15s of the movie. Were there efforts made to ensure there were no (within-movie) boundaries during this period? I was not sure whether these would be different (and if so, how different) from the event offset comparisons.

d) I was not sure how the "higher-than-average" and "lower-than-average" regions were defined, specifically. What is the average in this case?

e) Page 6, line 130: this phrasing was confusing to me: "this correlation was higher at boundaries than at non-boundaries (Figure 2A, blue arrows)." It made me think that the "blue arrows" in the parentheses may refer to the non-boundaries, but the blue arrows in the figure appear to be across-movie correlation for both boundary (left activation pattern pair in Figure 2A), and non-boundary (right pair Figure 2A) time points. Perhaps rephrase? Maybe "Figure 2A, a versus b blue arrows"?

f) Another phrasing question: "…, suggesting that the between-movie boundary pattern may reflect a cognitive state qualitatively different from the state elicited by event boundaries during movie watching (e.g., attentional engagement; Song et al., 2021)." I'm not quite sure what the authors mean by "e.g., attentional engagement". Can you explain further? Differences in attentional engagement across these two experience types, perhaps?

g) As may be clear from my public comment #2, the presentation of results in Figure 3D was hard for me to understand. Perhaps the authors could think of ways to plot this differently or spell out more explicitly the patterns of interest in the paper or legend to help readers understand. I would have loved more hand-holding for this analysis.

*Reviewer #2 (Recommendations for the authors):*

We really enjoyed this manuscript and had very few suggestions or requests for tweaks. We note these below:

1. We recognize that this is a short-format submission, but honestly, the 'Discussion' section of this paper just felt far too short. These are very interesting and thought-provoking results, which did not feel commensurate with the very small amount of text devoted to unpacking the findings and their implications. Note that we are not suggesting the authors did a poor job of constructing these two paragraphs, but rather we think that this is simply not enough space to really do justice to a broader conversation that can be had about these findings. We suggest (if possible, from an editorial standpoint) that a lengthier discussion could really solidify the contributions of this paper and make the connections to other studies and other ideas about event cognition and event memory more explicit. In particular, I would love to hear a more fleshed-out exploration of what this general cognitive state that may exist between events might look like, and what it might be doing.

2. The authors operationalized boundary periods as "…the first 15 seconds following the offset of each recalled movie, and the non-boundary periods…" as "…the 15 seconds in the middle of each recalled movie." Why was a window of 15 seconds chosen? While I do not have any reason to suspect the results would be wildly different if, say, a 10 or 20 second window was used, we think some justification or indication of why this particular parameter was chosen would be good to include.

3. Regarding Figure 2, we personally think that a different color gradient might be a better choice for the dark gray inflated brain image. This is a minor nit-pick, but the darker red colors do not show up against the dark sulcus portions of the map particularly well, and a different gradient might be preferable. This would be particularly true if individuals were to view this figure in grayscale, or in cases of color-blindness.

4. If so many individual parcels from the Schaefer atlas were examined, it is not clear why PCC and Precuneus were combined into a PMC ROI. While we do not particularly suspect that PCC and precuneus should look very different, we think a brief note of why these regions were combined and many others weren't would be good to include.

5. Regarding the correlations between the average between-movie boundary pattern and (1) between-movie boundaries, (2) moments of silence, (3) within-movie boundaries (i.e., the data displayed in Figure 4), we had to re-read this figure caption and Results section a few times to feel that we really grasped what comparisons were being made, and how these correlations were being conducted. We think this could stand to be tweaked to be clearer, with perhaps a conceptual diagram like the one in Figure 2A included to aid in understanding.

6. Following from the above point, we were initially not totally clear on the relevance or importance of the 'no sound' condition. Upon reading further into the Results, it made sense, but seeing as this figure is referenced well before this condition is expanded upon, we think this could be motivated a little earlier to mitigate unnecessary confusion. As of now, this set of analyses seems to be motivated after presentation of the data rather than before, and simply shifting this around could aid in understanding.

*Reviewer #3 (Recommendations for the authors):*

Specific suggestions the authors related to the points mentioned above:

1. To address the effect of similarity between encoding and recall due to the (lack of) perceptual input, it would be relevant to add an additional comparisons to the analyses as shown in figure 2: the similarity between middle segments during recall and boundary segments during encoding.

2. I wonder why the authors choose to focus on a long window of 15 seconds after stimulus onset rather than a shorter window. I would suggest investigating the similarity between the within-movie and between-movie event boundaries by focusing on a shorter window (e.g. 2 TRs). Additionally, I would propose running an analysis like the one in figure 3D for the within-movie and between-movie event boundaries to see how the (dis)similarity develops over time.

3. It is very unclear at this point whether the observed pattern of neural responses is related to the specific setup of the study or whether subtle shifts in context versus more intense transitions involve fundamentally different mechanism. I understand that this question probably cannot be answered with the current dataset, but I would appreciate some additional discussion related to this question, as well suggestions for how this could be clarified in future studies. I think the paper might be more suitable for a longer format, with a separate results and Discussion section. That would allow for a more extensive discussion for what the results might mean for our current understanding of how event segmentation works.

---

## [Author Response]

Essential revisions:1. Experiential similarity with the blank screen: How might the similarity of experience (black screen, no auditory input) across boundary periods be driving the analysis? Please consider this in a revision. Specifically, the authors should: (a) revise (perhaps correct, or at least clarify) statements in the text that seem to suggest consistent input across boundary and non-boundary periods (see comments from Reviewer #1 and #3 for more on this point). Moreover, (b) please perform additional analyses to control for this confound. Reviewers had specific suggestions for how the authors might perform these controls, but the authors are also welcome to carry out these controls in a different way, as they see fit.

We thank the reviewers for their comments and suggestions on the potential confounds due to similar visual and auditory input across boundary periods. First of all, we fully agree that boundary periods share perceptual and motoric features within each experimental phase, and these shared experiences likely contributed to higher within-phase pattern similarity for boundary periods compared to non-boundary periods. This issue was described in our original text but with insufficient clarity and prominence, for which we apologize. Indeed, this explains why more parcels showed significantly positive boundary pattern similarity *within* the recall phase than *across* encoding and recall phases (Figure 2B/C). We revised the Results section of the manuscript as below to make this point clearer:

p.5. “We observed a consistent boundary pattern, i.e., whenever participants transitioned from talking about one movie to the next, in several cortical parcels (Schaefer et al., 2018) including the DMN and auditory/motor areas (Figure 2B). Thus, the boundary patterns within the recall phase were likely to be driven by both shared low-level sensory/motor factors (e.g., breaks in recall speech generation) as well as cognitive states (e.g., memory retrieval) at recall boundaries.”

As the confound could not be resolved in the current dataset, the rest of our manuscript focused on the boundary pattern which was consistent *across* encoding and recall phases. This transient across-phase pattern similarity, observed specifically during boundary periods but not during non-boundary periods, is far less easily explained by shared perceptual or motoric features:

– Subjects viewed the black screen throughout the entire recall phase, so if the black screen at the boundaries during encoding was driving the boundary pattern, the similarity between encoding boundary patterns and recall non-boundary patterns should also be positive, but it was not (see Figure 3A/D).

– Likewise, subjects did not make any verbal response throughout the entire encoding phase, so if the absence of verbal responses at the boundaries during recall was responsible for the boundary pattern, the similarity between recall boundary patterns and encoding non-boundary patterns should also be positive-- but it was not.

We clarified this point by revising the ambiguous sentence in the Result section (“boundary and non-boundary periods were identical in terms of visual input during recall and speech generation during movie watching"), as suggested by Reviewer 1. The sentence was supposed to state two separate facts: (1) visual input was identical across boundary and non-boundary periods *during recall*, as subjects viewed a fixation dot on black screen throughout the recall phase, and (2) boundary and non-boundary periods were identical in terms of speech generation (i.e., no speech generated) *during movie watching*, as subjects did not make verbal responses at all throughout the encoding phase. We revised the text as below:

p.11. “Is the generalized between-movie boundary pattern driven by shared low-level perceptual or motoric factors rather than cognitive states? First, shared visual features at between-movie boundaries (i.e., black screen) cannot explain the transient, boundary-specific similarity between encoding and recall phases, because visual input was identical across boundary and non-boundary periods during recall (i.e., a fixation dot on black background). Indeed, encoding boundary patterns were more similar to recall boundary patterns than to recall non-boundary patterns in DMN areas, suggesting a limited contribution of shared visual input to the generalized boundary pattern (Figure 2—figure supplement 2). Likewise, the absence of verbal responses at boundaries cannot explain the boundary pattern generalized across encoding and recall phases, as no speech was generated throughout the entire encoding phase.”

In addition, we performed two new analyses to address the possible effects of shared input at boundaries:

First, following the suggestions of Reviewers 1 and 3, we performed a new analysis to control for the effects of shared visual input. Specifically, we directly compared the similarity between encoding and recall boundary patterns against the similarity between encoding boundary patterns and recall non-boundary patterns. Again, if the generalized boundary patterns that we observed were predominantly driven by shared low-level visual features, the similarity between encoding and recall boundary patterns would be approximately the same as the similarity between encoding boundary patterns and recall non-boundary patterns. However, we found that several cortical parcels, especially within the DMN, showed significantly greater encoding-recall boundary pattern similarity compared to the similarity between encoding boundary patterns and recall non-boundary patterns; the resulting maps were highly similar to the results reported in Figure 2C. Thus, the consistent boundary patterns observed in higher associative areas were not likely to be mainly due to low-level visual features shared across the experimental phases. We now report this analysis in Figure 2—figure supplement 2.

We revised the Results and Methods sections of the manuscript accordingly:

p.5. “First, shared visual features at between-movie boundaries (i.e., black screen) cannot explain the transient, boundary-specific similarity between encoding and recall phases, because visual input was identical across boundary and non-boundary periods during recall (i.e., a fixation dot on black background). Indeed, encoding boundary patterns were more similar to recall boundary patterns than to recall non-boundary patterns in DMN areas, suggesting a limited contribution of shared visual input to the generalized boundary pattern (Figure 2—figure supplement 2).”

p.22. “Testing the effect of visual features

Between-movie boundary periods during encoding and those during recall shared low-level visual features (i.e., mostly blank black screen). To test whether the similar visual features produced similar activation patterns at between-movie boundaries across phases, we performed a whole-brain pattern similarity analysis (Figure 2—figure supplement 2). For each subject and cortical parcel, we computed the mean boundary and non-boundary activation patterns for each movie, separately for encoding and recall. The boundary periods were defined as the first 15 seconds following the offset of each watched or recalled movie. The non-boundary periods were defined as the middle 15 seconds of each movie. Both boundary and non-boundary time windows were shifted forward by 4.5 seconds. We then computed Pearson correlations between encoding boundary patterns and recall boundary patterns across different movies, and averaged all the correlations. Likewise, we computed the average correlation between boundary patterns during encoding and non-boundary patterns during recall across different movies. A group-level two-tailed paired-samples t-test was performed to test whether the similarity between encoding and recall boundary patterns was greater than the similarity between encoding boundary patterns and recall non-boundary patterns, even though boundary and non-boundary patterns were visually identical during recall. The resulting whole-brain map was corrected for multiple comparisons across parcels using the Bonferroni method.”

Second, to further examine the effects of similar experiences at boundaries, we performed a new analysis (see Author response image 1) comparing the similarity between *boundary* patterns from *different* movies (Mb-diff) to the similarity between *non-boundary* patterns from the *same* movies during encoding (Mnb-same) in PMC. If the consistent boundary patterns were simply driven by similar external experiences during boundary periods, the boundary pattern similarity between different movies could not be higher than the non-boundary pattern similarity between the same movies; the external experience was “exactly” the same across the correlated non-boundary periods, because they were the same middle segments from the identical movie. Note that in this analysis, we had to compute between-subject pattern similarity (i.e., one subject’s pattern was correlated with the remaining subjects’ patterns) rather than within-subject pattern similarity, because there was only one non-boundary pattern for each movie and subject as each subject watched each movie only once. However, as shown in the left panel of Author response image 1, the boundary pattern similarity across different movies was higher than the non-boundary pattern similarity across the same movies. Moreover, even the boundary pattern similarity *across different movies and experimental phases* (MRb-diff) was numerically higher than the non-boundary pattern similarity across the same movies within the encoding phase (Mnb-same)(right panel). Together, these results suggest that shared cognitive features beyond audiovisual input contributed to the generalized boundary pattern. Because all other pattern-similarity analyses in the manuscript measured within-subject similarity, we opted not to report these results in the manuscript for consistency and to avoid readers’ confusion. However, we would be happy to include these results if the reviewers feel it will improve the manuscript.

**Author response image 1. sa2fig1:** 

2. Selection of temporal epochs for both boundary and non boundary time periods: All Reviewers were concerned about the long boundary period (15s) – they had questions including why it was defined this way, whether that decision matters, and how it might cloud interpretations of comparisons across these different time period (since, for example, we might expect boundary signatures to be more transient for within-movie boundaries than between-movie ones). The Reviewers have conferred on this issue and suggest the authors use a shorter temporal window to address this comment (rather than e.g., provide justification in the text) as suggested by Reviewer #3, given the expected transience of the univariate response.

Our choice of the boundary/non-boundary period time windows was primarily based on the time courses of boundary-related univariate responses, examined during the basic exploration of recall offset effects. As shown in Figure 1—figure supplement 1, boundary-related signals tended to last up to 30 seconds from recall offset, with the largest responses (deactivation) observed around approximately 15 seconds from the offset. Thus, we decided to use a time window of 15 seconds (shown as a red bar on the x axis) to include most of the time points that showed boundary responses.

In addition, as shown in Figure 3D, the time-time correlations between activation patterns around boundaries indicated that the generalized boundary activation pattern in PMC lasted until around 15 seconds from the movie/recall offset. That is, activation patterns remained stable for at least about 15 seconds from the offset. Thus, we aimed to generate clearer activation patterns to be used for analyses by taking the average of the stable patterns across time points that spanned the 15-s window.

We now clarify our decision for using a relatively long time window of 15 seconds in the Methods section as below:

p.18. “The boundary periods were the first 15 seconds following the offset of each recalled movie, and the non-boundary periods were the 15 seconds in the middle of each recalled movie. Both boundary and non-boundary period time windows were shifted forward by 4.5 seconds to account for the hemodynamic response delay. We used a relatively long 15-s duration for the boundary and non-boundary periods to capture most of the boundary-related signals during recall, based on exploratory analyses that examined the time courses of univariate boundary responses (Figure 1—figure supplement 1) and boundary-triggered activation patterns (Figure 3—figure supplement 4D).”

That said, we agree with the reviewers that 15 seconds (10 TRs) is a relatively long time window for pattern similarity analysis. As suggested by the reviewers, we re-analyzed the data using a shorter time window (3 TRs = 4.5 seconds) which is more conventional in pattern-based analyses, and obtained similar results. For example, we observed generalized boundary patterns consistent across different movie stimuli and experimental phases in the DMN areas. We now report these results in Figure 2—figure supplement 1 and Figure 3—figure supplement 3.

We revised the Results section and the Methods section of the manuscript accordingly:

p.7. “We observed similar results in the lateral parietal DMN sub-region (angular gyrus; Figure 3—figure supplement 2), as well as using shorter (4.5 s) time windows of boundary and non-boundary periods (Figure 2—figure supplement 1, Figure 3—figure supplement 3).”

p.19. “We also performed the same pattern similarity analysis in the PMC (Figure 3) and angular gyrus (Figure 3—figure supplement 2) ROIs, as done for an individual cortical parcel in the whole-brain analysis. In addition, we repeated the same analyses using shorter (4.5 seconds) boundary and non-boundary period time windows and obtained similar results (Figure 2—figure supplement 1, Figure 3—figure supplement 3). “

Finally, to test whether using a relatively longer/shorter time window would affect the similarity across the between-movie boundary pattern and the within-movie boundary pattern, we performed two extra analyses and reported the results in Figure 4—figure supplement 2.

First, we repeated the pattern similarity analysis comparing the ‘template’ between- and within-movie boundary patterns in PMC using a shorter (4.5 seconds) time window (Figure 4—figure supplement 2A). We replicated our original finding that there was no positive correlation across the two types of boundary patterns; the correlation was numerically negative, although it was not significantly different from zero.

Second, as Reviewer 3 suggested, we performed the time-time pattern correlation analysis comparing between-movie boundaries to within-movie boundaries in PMC (Figure 4—figure supplement 2B). This analysis examined whether the (dis)similarity between the two types of boundary patterns would change across time: namely, whether the within-movie boundary pattern was in fact similar (rather than dissimilar) to the between-boundary pattern in the short time window immediately following the boundary, but our long time window had caused us to miss the effect. However, we observed no effect of time windows as the patterns temporally unfolded; there was no significantly positive correlation across within- and between-movie boundary patterns in any of the time points following boundaries. Overall, these results support our original finding that within- and between-movie boundary patterns do not resemble each other.

We revised the Results section and the Methods section to report the new results:

p.9. “Within-movie and between-movie boundary patterns did not resemble each other, regardless of the specific time windows used to define the boundary periods (Figure 4—figure supplement 2).”

pp.21-22. “We also repeated the same pattern similarity analysis using shorter (4.5 s) time windows for the boundary periods, from 4.5 to 9 seconds following the within- or between-movie boundaries (Figure 4—figure supplement 2A).”

To explore the temporal unfolding of the similarity between the within- and between-movie boundary patterns, we additionally examined the between-phase TR-by-TR pattern similarity across individual time points around the boundaries (Figure 4—figure supplement 2B). For each subject, we extracted the PMC activation pattern time series from 30 seconds before to 60 seconds after 1) each within-movie event boundary during encoding and 2) the offset of each movie during recall. The time series were averaged across boundaries within each experimental phase. We then computed Pearson correlation coefficients across different time points in the activation pattern time series between the encoding and recall phases. Finally, we performed two-tailed one-sample *t*-tests against zero on each cell of the time-time correlation matrices from all subjects to identify the time points at which significantly positive or negative pattern correlations appeared. Bonferroni correction was applied to correct for multiple comparisons across cells.”

Additionally, as the similarity across between- and within-movie boundary patterns was only numerically negative when we used the shorter time window, we revised the following sentence in the abstract to deemphasize the “negative” correlation between the two.

p.2. “Surprisingly, the between-movie boundary patterns did not resemble patterns at boundaries between events within a movie.”

3. Concerns about within-run pattern similarity analysis: It would be important to know whether the results hold when limiting to only cross-run comparisons (see more from Reviewer #1). Moreover, please clarify in the revision how many runs there were for the recall task.

We first apologize for not clearly reporting how many scanning runs there were for the recall task. There were always two encoding runs for all subjects, but only a subset of subjects had two recall runs: 4 subjects had two recall runs, and the other subjects had one recall run. We wrote in our prior manuscript using the same dataset (Lee and Chen, 2021, bioRxiv, https://doi.org/10.1101/2021.04.24.4412874) that “in case subjects needed to take a break or the duration of the scanning run exceeded the scanner limit (35 minutes), we stopped the scan in the middle and started a new scanning run where subjects resumed from where they had stopped in the previous run. 4 of the 15 subjects included in the analysis had such a break within their spoken recall session.” We have revised the Methods section of the current manuscript as below to report the number of recall phase scanning runs.

p.17. “The recall phase consisted of two scanning runs in 4 of the 15 subjects included in the analysis. The other subjects had a single scanning run.”

Thus, as Reviewer #1 pointed out, the results of within-phase pattern similarity analyses (encoding-encoding and recall-recall) were partially derived from within-run pattern correlations. It is also true that correlating patterns within the same run could potentially lead to greater pattern similarity. However, we believe that temporal autocorrelation within a scanning run had little influence on our reported results, for the following reasons:

First of all, we did not simply show positive pattern correlations between boundary patterns; we compared them against pattern correlations between non-boundary patterns, which was also partially derived from within-run pattern correlations. Thus, any effect of within-run comparisons would influence both boundary and non-boundary conditions. Yet, we still observed much greater similarity across boundary patterns compared to non-boundary patterns. Moreover, the main focus of our study was the generalized boundary pattern consistent across encoding and recall phases, which was computed purely across different scanning runs.

Finally, to directly test the effect of computing within-run pattern correlations, we performed a new pattern similarity analysis comparing within-run pattern similarity and between-run pattern similarity using the PMC encoding phase data. Recall data were not considered, as there were only 4 subjects who had two separate scanning runs. As shown in Author response image 2, we computed the mean within-run, between-movie pattern similarity across boundary patterns (Within-run Boundary) and across non-boundary patterns (Within-run Nonboundary). Likewise, we computed the mean between-run, between-movie pattern similarity across boundary patterns (Between-run Boundary) and across non-boundary patterns (Between-run Nonboundary).

The results show that between-movie pattern similarity was greater in the boundary than non-boundary conditions regardless of whether the pattern similarity was measured within the same run or between different runs (*t*s > 10, *p*s <.001). Moreover, the overall boundary pattern similarity did not differ across within-run and between-run correlations (i.e., Within-run Boundary vs. Between-run Boundary; *t*(14) = 1.21, *p* = .25). Thus, the generalized boundary pattern we reported in our manuscript was not likely to be primarily driven by potential confounds associated with within-run comparisons. This also suggests that although we observed more widespread Recall-Recall than Encoding-Recall similarity in Figure 2B-C, it is likely to be due to shared sensory/motor factors within the recall phase as we noted in our manuscript (“the boundary patterns within the recall phase were likely to be driven by both shared low-level sensory/motor factors (e.g., breaks in recall speech generation) as well as cognitive states (e.g., memory retrieval) at recall boundaries.”), rather than the effect of temporal autocorrelations.

4. Providing more background for the focus on PMC: What was the reason for the focus on PMC? Reviewer #1 was not sure whether the results were specific to the PMC, and/or whether and why this was an a priori ROI. Are the operations thought to be unique to the PMC, or shared with other DMN regions? Also, Reviewer #2 was not sure why PCC and precuneus were combined to make PMC rather than considered separately. Please provide more on the logic behind these choices.

We focused on PMC primarily because PMC was the area that showed the strongest content- and task-general boundary patterns in the exploratory whole-brain analysis (Figure 2C). We clarified this reasoning behind our choice of PMC as the major region of interest in the manuscript as below:

p.7. “To what extent is the internally-driven boundary pattern, measured during recall, similar to patterns observed at boundaries during encoding? To test this, we again computed between-movie pattern similarity for all cortical parcels in the brain, but now across the encoding and recall phases (Figure 2A, red arrows). We found that DMN areas showed a consistent boundary pattern across task phases (encoding and recall) and across movies (Figure 2C). Again, no cortical area showed negative correlations between boundary patterns or greater correlations for non-boundary compared to boundary patterns. Among the DMN areas, the posterior medial cortex (PMC) showed the most consistent boundary patterns; thus, we next examined the phenomenon in more detail specifically in PMC.”

We also had a priori speculation that PMC is likely to be involved in processing transitions between situations or events, based on prior studies demonstrating that the region accumulates information over long timescales (e.g., Hasson et al., 2015) and represents relatively abstract situation-level information (e.g., Ranganath and Ritchey, 2012; Chen et al., 2017). However, this does not necessarily mean that these operations are unique to PMC. Indeed, we observed generalized boundary patterns in other DMN areas as well, especially in the angular gyrus, as demonstrated in the whole-brain map in Figure 3C. We therefore performed pattern similarity analysis targeting the angular gyrus, and obtained results highly similar to those obtained from PMC. The angular gyrus results are reported in Figure 3—figure supplement 2 and mentioned in the main text as below:

p.7. “Individual subjects’ activation maps visualize the similarity between boundary patterns during encoding and recall (Figure 3B, Figure 3—figure supplement 1). We observed similar results in the lateral parietal DMN sub-region (angular gyrus; Figure 3—figure supplement 2), as well as using shorter (4.5 s) time windows of boundary and non-boundary periods (Figure 2—figure supplement 1, Figure 3—figure supplement 3).”

We created a single PMC ROI mask by combining the PCC and precuneus to be consistent with our prior study that analyzed the same region of interest (Lee and Chen, 2021). Our decision to use PMC including both the PCC and precuneus in our current and prior studies was based on the atlas of functional networks widely used in the field (Schaefer et al., 2018). In the Schaefer atlas, the two regions together form the posterior medial sub-region of the default network, and the parcels that we combined to create the PMC ROI all fell within the default network. In addition, we observed the content- and task-general boundary pattern in the larger PMC area in our whole-brain analysis (Figure 2C), and thus we opted to use a single large PMC mask for our ROI analysis rather than breaking the continuous area into several sub-regions. As Reviewer 2 mentioned in their Recommendations #4, it is unlikely that the PCC and precuneus would show qualitatively different results, because the whole-brain analysis using smaller parcels already demonstrated that the PCC and precuneus as well as their fine-grained sub-areas all showed boundary patterns consistent across experimental phases. We revised the Methods section of the manuscript as below to explain our reasons for combining the two regions:

p.17. “For region-of-interest analyses, we defined the bilateral posterior-medial cortex (PMC) by combining the parcels corresponding to the precuneus and posterior cingulate cortex within Default Network A as in our prior study (Lee and Chen, 2021). The precuneus and posterior cingulate cortex together spanned the area that showed the strongest content-and task-general boundary patterns in the whole-brain analysis (Figure 3C). The bilateral angular gyrus ROI consisted of the parcels corresponding to the inferior parietal cortex within Default Network A, B and C.”

5. Univariate responses to boundaries: The description of the results in Figure 1B do not seem to match with the results show in the figure. There seem to be widespread transient changes and a larger number of vertices that show a decline in BOLD activity after a boundary. In addition, there seems to be a strong positive BOLD response throughout the default mode network. Please clarify.

We apologize for not being clear in describing our whole-brain univariate activation results. In the original version of the manuscript, we described the changes in univariate activity as “limited” because there were only a few vertices that showed *significant* effects after multiple comparisons correction (shown as small white dots in the original version of Figure 1B). Note that the red-blue color gradient in Figure 1B indicates the level of *unthresholded* responses, shown to provide a more complete picture of univariate responses at between-movie boundaries. However, we fully agree with the reviewers that there were widespread responses across the entire cortex at between-movie boundaries, although they did not reach statistical significance after Bonferroni correction. In addition, applying Bonferroni correction to 81924 vertices might be an overly conservative correction. Thus, we applied the Benjamini-Hochberg FDR correction instead of Bonferroni and revised Figure 1B.

By switching to FDR correction, we were able to identify more widespread areas that showed significant changes in activation following between-movie boundaries (including both increase and decrease in activation), consistent with Reviewer 3’s observation. We believe that this revised result provides a better description of the neural phenomena associated with between-movie boundaries, compared to the previous result (i.e., significant BOLD changes in only a few vertices) based on Bonferroni correction. We revised the Results and the Methods accordingly:

p.5. “We first examined whether internally-driven boundaries evoke changes in blood oxygen level-dependent (BOLD) signals during recall. We observed transient changes in activation at the boundaries between recalled movies in widespread cortical regions (Figure 1—video 1; see Figure 1—figure supplement 1 for activation time courses). A whole-brain analysis with multiple comparisons correction revealed that the mean activation of boundary periods (15 seconds following the offset of each movie) was generally lower than that of non-boundary periods (middle 15 seconds within each movie) in multiple areas including the motor, auditory, and inferior parietal cortices, although a smaller number of regions showed higher activation during non-boundary periods (Figure 1B).”

p.18. “The Benjamini-Hochberg procedure (False Discovery Rate *q* <.05) was applied to correct for multiple comparisons across vertices on the resulting whole-brain statistical parametric map.”

In addition, we now present activation time courses around between-movie boundaries from three regions of interest used in the study (posterior medial cortex, angular gyrus, and auditory cortex) in Figure 1—figure supplement 1. The activation time courses show transient increases in activation followed by longer-lasting decreases in activation at between-movie boundaries during recall. Thus, although the whole-brain maps in Figure 1B show positive responses in a subset of DMN areas, both positive responses and (perhaps more predominantly) negative responses are present in the DMN following boundaries. Figure 1—video 1 also clearly demonstrates the whole-brain BOLD time course.

6. Paper format: Reviewers felt the paper would be better suited for a longer format article, with separate Results and Discussion sections. That would allow for a more extensive discussion (see following points) of what the results might mean for our current understanding of how event segmentation works.

We agree with the reviewers that the manuscript can be improved by separating the Results and Discussion sections and adding more in-depth treatment of the findings. Following the reviewers’ suggestions, we divided Results and Discussion into two separate sections and substantially expanded the Discussion (see our responses for Comments 7-8 below). We greatly appreciate the opportunity to expand on our ideas in a longer format.

7. Discussion: Reviewers felt the Discussion was too short, and felt the authors' interesting and thought-providing findings warranted a more extensive unpacking. Reviewers would like to see a more fleshed-out exploration of what this general cognitive state that may exist between events might look like, and what it might be doing.

We thank the reviewers for their suggestions. We expanded the discussion on the potential nature of the general boundary state in the revised Discussion section as below. We also added extended text relating our results to existing empirical and theoretical work. Please see the revised manuscript for the remaining parts of the Discussion section.

p.15. “What is the cognitive state that is generalized across internal- and external boundaries between completely different contexts, but distinct from the state evoked by boundaries within the same context? We speculate that the between-movie boundary state may be a temporary “relay” state that occurs when no one mental model wins the competition to receive full attentional focus following the flushing of the prior mental context. Namely, when one major mental context switches to another, the brain may pass through a transient off-focus (Mittner et al., 2016) or mind-blanking (Mortaheb et al., 2022; Ward and Wegner, 2013) state which is distinct from both processing external stimuli (e.g., movie watching) and engaging in internal thoughts (e.g., memory recall). This account may also explain the difference between within- vs. between-movie boundary patterns: in terms of attentional fluctuation (Jayakumar et al., 2022; Song, Finn, et al., 2021), external attention is enhanced at within-movie event boundaries (Pradhan and Kumar, 2021; Zacks et al., 2007), whereas the relay state is associated with lapses in attention (deBettencourt et al., 2018; Esterman et al., 2014). An alternative, but not mutually exclusive, possibility is that the boundary state involves the recruitment of cognitive control to resolve the competition between mental contexts. This idea is based on the observation that the areas showing relatively higher activation at between-movie boundaries overlap with the frontoparietal control network (FPCN; Vincent et al., 2008) both during encoding and recall (Figure 1B, Figure 1-video 2). As the FPCN is interdigitated with the DMN and other nearby areas within individual subjects (Braga and Buckner, 2017), relative activation of the FPCN may create the stereotyped boundary pattern in higher associative cortices. It is also noteworthy that both of these candidate cognitive states are triggered not by the onset but by the offset of a mental context; the onset would rather signal the resolution of competition between mental contexts, hence the end of those states. This dovetails with our results showing that the generalized boundary pattern appears well before movie onsets, suggesting a major contribution of offset responses.”

8. Also to consider in the expanded the Discussion: It is very unclear at this point whether the observed pattern of neural responses is related to the specific setup of the study or whether subtle shifts in context versus more intense transitions involve fundamentally different mechanism. I understand that this question probably cannot be answered with the current dataset, but I would appreciate some additional discussion related to this question, as well suggestions for how this could be clarified in future studies.

Following the Reviewers’ suggestions, we now provide more in-depth discussion on the within- vs. between-movie boundary distinction (i.e., subtle vs. intense transitions) in the revised Discussion section as below:

p.14. “Although the boundary-related PMC activation patterns were consistent across internally- and externally-driven boundaries, they did not generalize across within- and between-movie boundaries. Relatedly, a recent human neurophysiological study (Zheng et al., 2022) reported that medial temporal cortex neurons distinguished within- and between-movie boundaries while subjects were watching short video clips; some neurons responded only to between-movie boundaries, whereas a separate group of neurons responded to both types of boundaries. These findings may be in line with the view that event boundaries have a hierarchical structure, with different brain areas along the information pathway reflecting different levels of boundaries, from fine-grained sensory transitions to coarse-grained situational transitions (Baldassano et al., 2017; Chang et al., 2021; Geerligs et al., 2021). However, it is still puzzling that within- and between-movie boundaries in our study produced qualitatively distinct neural patterns within a highest-order area (PMC), even though both categories consisted of prominent boundaries between situations spanning tens of seconds to several minutes. What are the crucial differences between the two levels of boundaries? One important factor might be the presence or absence of inter-event connections. Even the most salient within-movie boundaries still demand some integration of information across events, as the events are semantically or causally related, and ultimately constitute a single coherent narrative (Lee and Chen, 2021; Song, Park, et al., 2021). In contrast, an entire cluster of related events, or the narrative as a whole, might be completely “flushed” at between-movie boundaries; this difference could induce distinct cognitive states at the two levels of boundaries, giving rise to different PMC patterns.”

In addition, in the last paragraph of the Discussion section, we suggest a future study for testing whether our findings would be generalized in more realistic and unconstrained settings:

pp.15-16. “In conclusion, we found that internally-driven boundaries between memories produce a stereotyped activation pattern in the DMN, potentially reflecting a unique cognitive state associated with the flushing and updating of mental contexts. By demonstrating stimulus-independent event segmentation during continuous and naturalistic recall, our study bridges the gap between the fields of event segmentation and spontaneous internal thoughts (also see Tseng and Poppenk, 2020). Without any task demands or external constraints, the mind constantly shifts between different internal contexts (Raffaelli et al., 2021; Sripada and Taxali, 2020). What are the characteristics of neural responses to different types of spontaneous mental context boundaries (e.g., between two different memories, between external attention and future thinking)? Is the boundary pattern observed in the current study further generalizable to mental context transitions even more stark than between-movie transitions in our experiment? Are there specific neural signatures that predict subsequent thought transitions? Future work will explore answers to these questions by employing neuroimaging methods with behavioral paradigms that explicitly and continuously track the unconstrained flow of thoughts in naturalistic settings.”

9. Figure 4 results presentation: Regarding the correlations between the average between-movie boundary pattern and (1) between-movie boundaries, (2) moments of silence, (3) within-movie boundaries (i.e., the data displayed in Figure 4), we had to re-read this figure caption and Results section a few times to feel that we really grasped what comparisons were being made, and how these correlations were being conducted. We think this could stand to be tweaked to be clearer, with perhaps a conceptual diagram like the one in Figure 2A included to aid in understanding.

We apologize for the ambiguity in visualizing our results in Figure 4. To help readers understand the figure, we divided Figure 4 into two different figures: Figure 4 reporting the between-movie vs. within-movie boundary pattern comparison, and Figure 5 reporting control analyses regarding the silence at boundaries (also see our response to Comment #10). In the revised Figure 4 we added a new panel explaining the analysis method with a detailed caption. We used a color scheme that connects the specific comparisons depicted in the analysis schematic (Figure 4A) and the conditions reported in the result graph (Figure 4B). Note that we now report results from PMC only, as the within- vs. between-movie boundary comparison in the auditory cortex was not the focus of the analysis and was not discussed in the text.

We also revised the Results section and the Methods section as below. We report the correlation between the between-movie and within-movie boundary patterns within the encoding phase (in addition to the cross-phase correlation) to provide more complete results from the analysis.

p.9. “The within-movie event boundary pattern was also negatively correlated with the encoding phase between-movie boundary pattern (*t*(14) = 7.31, *p* <.001, Cohen’s *d*_z_ = 1.89, 95% CI = [-.44, -.24]).”

p.21. “For each subject, we generated the mean within-movie event boundary pattern of PMC by averaging activation patterns from 4.5 to 19.5 seconds following each of the 15 event boundaries during encoding. The patterns were first averaged across all time points within each boundary period time window and then across different boundaries. Likewise, the mean between-movie boundary pattern was generated by averaging all activation patterns from 4.5 to 19.5 seconds following the offset of each movie during encoding or recall. We then computed a Pearson correlation coefficient across the mean within-movie event boundary pattern and the mean encoding or recall between-movie boundary pattern.”

To report the ‘No sound’ condition from PMC and the auditory cortex in the previous version of Figure 4, we created Figure 5. We additionally included a schematic of the analysis to help readers understand the analysis.

10. Clarifying no-sound condition: Following from the above point, we were initially not totally clear on the relevance or importance of the 'no sound' condition. Upon reading further into the Results, it made sense, but seeing as this figure is referenced well before this condition is expanded upon, we think this could be motivated a little earlier to mitigate unnecessary confusion. As of now, this set of analyses seems to be motivated after presentation of the data rather than before, and simply shifting this around could aid in understanding.

We thank the reviewers for their suggestion. As we wrote in our response to the comment #9 above, we divided Figure 4 into two so that the revised Figure 4 shows the comparison between within- and between-movie boundary patterns only, and the new Figure 5 shows the control analysis corresponding to the ‘no sound’ condition in the original Figure 4. We now present Figure 5 after the paragraph describing the control analyses, so that there is no confusion due to the order in which the results and figures are presented.

11. Figure 3 results presentation: I was not sure I fully understood the offset vs. onset yoking analysis-both how it was performed, and how the conclusions followed from the results. First, I was a bit confused about how the difference in delay duration between movies at encoding (6s) versus at recall (9.3s on average, but variable; see also comment #5) would play into this and whether those are meaningful time points to display on the Figure 3D charts that might help the reader interpret those findings.

We apologize for not being clearer in describing the analysis methods and conclusion of the offset vs. onset analysis. Regarding the delay duration, we first agree with the reviewer that displaying the delay duration between movie offsets and onsets on the time-time correlation matrices will facilitate interpretation of the results. Unlike in many prior studies where the offset of an event is identical to the onset of the following event (e.g., Baldassano et al., 2017), movie offsets and onsets were temporally separated in our study. This is an important point that allowed us to distinguish the temporal emergence of the boundary pattern following movie offsets versus onsets. Thus, we now indicate the delay between onsets and offsets, on Figure 3D and Figure 3—figure supplements 2 and 4, by marking the onsets/offsets of the following/preceding movies with dotted lines.

As for whether/how the difference in the delay duration between encoding and recall phases affected the offset vs. onset analysis results, we believe that our reported results and interpretation of the offset vs. onset analysis were not systematically influenced by the “longer” delay duration during recall. As the reviewer pointed out, different delay duration can affect activation patterns around between-movie boundaries; if the delay is short, there would be a greater chance that offset responses to one movie and onset responses to the following movie are mixed together. Thus, if we assume that the offset and onset patterns are not identical to each other, mixed responses due to shorter delay duration during encoding may lower the encoding-recall pattern similarity around boundaries time-locked to either onsets or offsets. However, this cannot explain why positive encoding-recall pattern correlations appear “before” movie onsets when the activation pattern time series were time-locked to the onsets, which is the main finding of the offset vs. onset analysis. Instead, if the offset and onset patterns were mixed together during encoding (i.e., relatively early periods of encoding onset responses are actually the mixture of onset and offset responses), one would expect that positive encoding-recall pattern correlations would appear much later following onsets. Therefore, the current results can be explained by either assuming that 1) the boundary patterns are predominantly driven by offsets, or 2) the boundary responses arise following both offsets as well as onsets, and their associated activation patterns are identical; both of these accounts reject the possibility that the boundary patterns were primarily driven by movie onsets. Moreover, the effect of mixed patterns would be identical regardless of whether the delay duration is shorter during encoding or during recall. Likewise, the “greater variability” in delay duration during recall is also not likely to systematically bias the offset vs. onset difference, as it would equally affect the offset-locked encoding-recall pattern similarity and onset-locked encoding-recall pattern similarity.

Second, the authors state that this analysis shows the boundary patterns were driven by offset (more than onset) responses, but I was not sure what aspect of the results led to that conclusion. Can the authors say more about the evidence supporting this conclusion? It looks to me like there are strong correlations that emerge both after offset and onset (i.e., just above and to the right of the origin there are numerous time points with positive [red] correlations). Perhaps it is because the red positive correlations start earlier, prior to the recall itself, when yoked to the onset, but I am not sure why this means it is related to offset and not some preparation for the onset of recall (see also comment #5).

As the reviewer mentioned, our conclusion that offset responses have significantly contributed to the generalized boundary pattern is based on the result shown in Figure 3D (left panel): positive encoding-recall pattern correlations arose several seconds before the onset of a movie. Especially considering hemodynamic response delay (i.e., the responses actually occurred even earlier), it is unlikely that the boundary pattern arose as a “reaction” to movie onsets only. That said, we acknowledge that our results do not completely rule out the possibility that movie onsets produced similar boundary patterns as the offset patterns, because on average the temporal gaps between onsets and offsets were not large enough to perfectly distinguish onset and offset signals. Thus, we have revised the Results section as below to reframe the question being addressed (“is it the onset effect?” rather than “is it the onset effect or the offset effect?”) and clarify our interpretation of the results:

p.7. “Is the generalized boundary pattern evoked by the onset of a movie, rather than the offset? We examined this question by comparing the temporal emergence of the generalized boundary pattern following movie offsets versus onsets (Figure 3D); note that the offset of a movie was temporally separated from the onset of the following movie during both encoding and recall (see Figure 1A). Specifically, we extracted the mean time series of PMC activation patterns around between-movie boundaries, time-locked to either the onset or offset of each watched or recalled movie. We then computed between-phase (encoding-recall) pattern similarity across the individual time points of the activation pattern time series. We found that significantly positive between-phase correlations emerged well before the encoding and recall onsets (Figure 3D, left panel), starting from 4.5 seconds following the offsets of the preceding watched or recalled movie (Figure 3D, right panel). Thus, boundary patterns were not exclusively triggered by movie onsets; it is likely that offset responses significantly contributed to the boundary patterns.”

Regarding the possibility that the boundary pattern that comes right before a movie onset is related to preparatory responses to the recall onset rather than a response to the offset of the preceding recall, we first agree that there might be preparatory cognitive processes before recall speech onsets associated with covert memory retrieval of the upcoming movie. However, as the reviewer wrote, “during recall, it seems as though the participant would be bringing to mind memories of the upcoming movie B they are about to recall, while there is no way for participants to anticipate anything specific about the upcoming movie during encoding.” Thus, the anticipatory recall processing would lead to more dissimilar cognitive states between encoding and recall phases; that is, it would decrease, rather than increase, the encoding-recall pattern similarity right before the movie/recall onset. Therefore, it is unlikely to be the main factor driving the positive encoding-retrieval pattern correlations that precede onsets. The generalized boundary pattern arises *despite* the preparatory responses, and the more parsimonious explanation is that offset responses significantly contributed to the boundary pattern.

Also, is it interesting or meaningful that the patterns seem more compressed at recall than at encoding (i.e., the outlined red areas are skinnier than they are tall)? Please clarify.

The reviewer’s observation is correct: boundary patterns were stronger and lasted longer during encoding than during recall, which was also observed in univariate activation responses (e.g., Figure 1—videos 1-2). This tendency is more clearly depicted in the within-phase time-time pattern correlation matrices in Figure 3—figure supplement 4. Our speculation is that boundary patterns lasted longer during encoding partly because the between-movie boundaries were more salient during encoding, as they accompanied both external and internal context changes, whereas recall boundaries accompanied internal context changes only. In addition, compared to self-generated boundaries between already stored memories during recall, boundaries between previously unseen movies during encoding are likely to be more unpredictable and may require building a completely new mental context for the post-boundary movie. These characteristics may prolong the boundary (“relay”) cognitive state during encoding. We now briefly discuss the differential boundary pattern duration between encoding and recall in the Figure 3—figure supplement 4 caption.

12. Reasoning behind masking decisions: I am not sure the reason for masking Figure 2B and C with the a>0 and c>0 maps. First of all, it seems as though in RSA the actual correlation being positive or negative is not terribly meaningful and can depend on preprocessing decisions, etc. In addition to that potential issue though I'm also just generally interested in more understanding the logic behind this decision. Can the authors explain that and include it in the main paper? Were there any regions that showed for example a<b, or a<0 and if so, why were these not considered?

We apologize for not being clear about the reasoning behind masking the boundary > non-boundary map with the boundary > 0 map. As the reviewer mentioned, raw correlation values are generally not very meaningful in pattern similarity analysis; more important are differences between conditions of interest. However, the current study aimed to show the spatial similarity between boundary patterns across different movies/tasks, such as shown in the individual subjects’ maps in Figure 3B, and thus we wanted to confirm that the correlations between patterns were at least greater than zero. At the same time, to make sure that the consistent spatial patterns were associated with boundaries and not driven by factors unrelated to fluctuating cognitive states/situations (e.g., blood vessel shape/location), we tested whether the boundary pattern similarity was greater than the non-boundary pattern similarity. Finally, by masking the boundary > non-boundary map with the boundary > 0 map, we aimed to identify the conjunction areas that satisfied both criteria; note that masking the boundary > 0 map with the boundary > non-boundary map would indicate the same areas. We revised the Methods section of the manuscript as below to explain our reasoning:

p.19. “Finally, we identified parcels that showed significant effects in both tests after the correction, by masking the areas that showed higher pattern similarity for the boundary than non-boundary conditions with the areas that showed overall positive similarity between boundary patterns (Figure 2B). Thus, the identified parcels showed spatially similar activation patterns across different movies at recall boundaries, and the patterns were specifically associated with boundary periods only.”

In Author response image 3 we also show the boundary > non-boundary maps and the boundary > 0 maps without masking, each corrected for multiple comparisons. The boundary > non-boundary maps and the boundary > 0 maps largely overlapped with each other in both Recall-Recall pattern similarity and Encoding-Recall pattern similarity; thus, the masking procedure actually had negligible effects on the masked whole-brain maps presented in Figure 2. We opted not to show these unmasked maps in the manuscript as they are redundant to the maps presented in Figure 2 and its supplements. However, we would be happy to include them if the reviewers feel it will improve the paper.

**Author response image 3. sa2fig3:** 

Finally, as shown in Author response image 3, no cortical parcel showed significantly negative pattern similarity between boundary patterns in both Recall-Recall pattern similarity and Encoding-Recall pattern similarity. Likewise, no cortical parcel showed significantly lower pattern similarity between boundary patterns compared to the similarity between non-boundary patterns. We now report these findings in the text as below:

p.5. “Thus, the boundary patterns within the recall phase were likely to be driven by both shared low-level sensory/motor factors (e.g., breaks in recall speech generation) as well as cognitive states (e.g., memory retrieval) at recall boundaries. No cortical parcel showed significantly negative correlations between boundary patterns or greater correlations in the non-boundary compared to boundary conditions.”

p.7. “To test this, we again computed between-movie pattern similarity for all cortical parcels in the brain, but now across the encoding and recall phases (Figure 2A, red arrows). We found that DMN areas showed a consistent boundary pattern across task phases (encoding and recall) and across movies (Figure 2C). Again, no cortical parcel showed negative correlations between boundary patterns or greater correlations in the non-boundary condition.”

13. Clarity suggestions: The Reviewers also had several other suggestions that might increase the clarity of results presentation and/or depictions. The authors should please consider these suggestions and implement if they see fit.

We thank the reviewers for providing valuable suggestions that helped us improve the clarity of the manuscript. We carefully addressed all suggestions and revised the text, figures, and figure supplements accordingly. Please see our responses to individual comments from the “Recommendations for the authors” by each reviewer.

Reviewer #1 (Recommendations for the authors):1. I had a couple more specific comments and ideas/suggestions:a) Regarding p. 6, line 218-219: "boundary and non-boundary periods were identical in terms of visual input during recall and speech generation during movie watching": Please clarify this phrasing. I thought boundary periods would be a blank screen, and non-boundary would be a movie playing during encoding and have verbal recall happening during recall. How is it true that they are identical in terms of visual input?

[Copied from Essential Revisions 1] We apologize for the ambiguous phrasing. The sentence was supposed to state two separate facts: (1) visual input was identical across boundary and non-boundary periods *during recall*, as subjects viewed a fixation dot on black screen throughout the recall phase, and (2) boundary and non-boundary periods were identical in terms of speech generation (i.e., no speech generated) *during movie watching*, as subjects did not make verbal responses at all throughout the encoding phase. We revised the text as below:

p.11. “Is the generalized between-movie boundary pattern driven by shared low-level perceptual or motoric factors rather than cognitive states? First, shared visual features at between-movie boundaries (i.e., black screen) cannot explain the transient, boundary-specific similarity between encoding and recall phases, because visual input was identical across boundary and non-boundary periods during recall (i.e., a fixation dot on black background). Indeed, encoding boundary patterns were more similar to recall boundary patterns than to recall non-boundary patterns in DMN areas, suggesting a limited contribution of shared visual input to the generalized boundary pattern (Figure 2—figure supplement 2). Likewise, the absence of verbal responses at boundaries cannot explain the boundary pattern generalized across encoding and recall phases, as no speech was generated throughout the entire encoding phase.”

b) For overcoming the concern about visual input, I am wondering if perhaps the authors might be able to leverage the title screen before the very first movie in a run as a control, since this is not really a "boundary"? Alternatively, perhaps duration of blank screen viewing could be useful as a control since if what this is picking up on is blank-screne-ness, maybe it would increase with more time spent looking at the blank screen (?).

We appreciate the reviewer’s suggestion for potential control analyses. However, unfortunately, the suggested analyses would not be able to provide an appropriate control for the effects of visual input considering the design of the current study. First, there was a short (39 s) introductory cartoon played at the beginning of each encoding scanning run (see ‘Stimuli’ in the Materials and methods section), thus the title scene before the first movie of a run was also a boundary between the cartoon introduction and the movie. As for the duration of blank screen viewing, the title scene duration was largely consistent across all movies, with minimal variability depending on whether the movie clips had extra black screen periods (usually < 3 s) at the beginning or end of the clips. Thus, we think the suggested analysis of relating the boundary pattern to the duration of blank screen viewing would not be sensitive enough to control for the effects of visual input.

That said, we believe that similar visual input (e.g., mostly blank black screen) is unlikely to produce the boundary pattern generalizable across encoding and recall, because the boundary pattern was transient (as shown in the time-time correlation matrices in Figure 3D) whereas visual input was consistent throughout the recall phase. [Copied from Essential Revisions 1] We confirmed this by performing an analysis directly comparing the similarity between encoding and recall boundary patterns against the similarity between encoding boundary patterns and recall non-boundary patterns. If the generalized boundary patterns that we observed were predominantly driven by shared low-level visual features, the similarity between encoding and recall boundary patterns would be approximately the same as the similarity between encoding boundary patterns and recall non-boundary patterns. However, we found that several cortical parcels, especially within the DMN, showed significantly greater encoding-recall boundary pattern similarity compared to the similarity between encoding boundary patterns and recall non-boundary patterns; the resulting maps were highly similar to the results reported in Figure 2C. Thus, the consistent boundary patterns observed in higher associative areas were not likely to be mainly due to low-level visual features shared across the experimental phases.

We revised the Results and Methods sections of the manuscript accordingly.

2. I have a number of clarification questions-these are mainly methodological choices that I simply didn't understand, and felt that further explanation or justification for these decisions should be included in the paper:a) What was the reason for the focus on PMC? I was not sure whether the results were specific to the PMC, and/or whether and why this was an a priori ROI. Are the operations thought to be unique to the PMC, or shared with other DMN regions? More logic behind this choice would be very helpful.

[Copied from Essential Revisions 4] We focused on PMC primarily because PMC was the area that showed the strongest content- and task-general boundary patterns in the exploratory whole-brain analysis (Figure 2C). We clarified this reasoning behind our choice of PMC as the major region of interest in the manuscript.

b) The run structure was a bit unclear to me. Sorry if I missed this, but I saw there were two encoding scans but could not work out how many recall scans there were.

[Copied from Essential Revisions 3] We apologize for not clearly reporting how many scanning runs there were for the recall task. There were always two encoding runs for all subjects, but only a subset of subjects had two recall runs: 4 subjects had two recall runs, and the other subjects had one recall run. We wrote in our prior manuscript using the same dataset (Lee and Chen, 2021, bioRxiv, https://doi.org/10.1101/2021.04.24.4412874) that “in case subjects needed to take a break or the duration of the scanning run exceeded the scanner limit (35 minutes), we stopped the scan in the middle and started a new scanning run where subjects resumed from where they had stopped in the previous run. 4 of the 15 subjects included in the analysis had such a break within their spoken recall session.” We have revised the Methods section of the current manuscript as below to report the number of recall phase scanning runs.

p.17. “The recall phase consisted of two scanning runs in 4 of the 15 subjects included in the analysis. The other subjects had a single scanning run.”

c) For the non-boundary comparisons, this was described as being the middle 15s of the movie. Were there efforts made to ensure there were no (within-movie) boundaries during this period? I was not sure whether these would be different (and if so, how different) from the event offset comparisons.

We thank the reviewer for raising this important point. Two of the 15-s non-boundary (middle) periods partially overlapped with within-movie event boundary periods by 3 TRs and 9 TRs, respectively. However, for our initial main analyses comparing between-movie boundary periods and non-boundary periods, we opted not to exclude or adjust the non-boundary periods that overlapped with the within-movie boundary periods. There were two reasons for this decision:

– First, within-movie boundaries were not of a priori interest. The focus of our study was internally-generated boundaries during recall; while between-movie boundaries were easy to identify during recall for all subjects, within-movie event boundaries often could not be reliably defined during recall because it was common for subjects to summarize or combine several events, and subjects varied in how they did so. This was the primary reason that we chose to examine between-movie boundaries rather than within-movie event boundaries.

– In addition, it is very difficult to completely avoid any event boundary when selecting a non-boundary time window within a movie. Although we included only the strongest within-movie event boundaries in our analysis to maximize effects when comparing within vs. between-movie boundaries, within-movie event boundaries are easily perceived at much finer-grained levels, occurring every few seconds. These finer-grained event boundaries could also be perceived at different moments across subjects. In other words, regardless of how we define non-boundary periods, there are almost always some possibilities of including these weaker within-event boundaries.

For these reasons, we chose not to consider the within-movie event boundaries in our primary boundary vs. non-boundary comparison; and thus it was reasonable for us to define non-boundary periods as the time windows farthest away from any between-movie boundaries (i.e., middle segments) rather than identifying and avoiding all possible within-movie boundaries. This choice also allowed us to define non-boundary patterns in all 10 movies and keep the non-boundary period definition consistent across movies.

That said, we performed extra analyses to confirm that including the two non-boundary periods that partially overlapped with within-movie event boundary periods did not influence the results in our study. Specifically, we computed the PMC pattern similarity between non-boundary patterns from different movies within the encoding phase (i.e., encoding-encoding non-boundary pattern similarity), and across encoding and recall phases (i.e., encoding-recall non-boundary pattern similarity), including or excluding the two periods overlapping with within-movie boundary periods. Likewise, we also computed the between-movie PMC pattern similarity between non-boundary and movie-offset patterns, within the encoding phase (i.e., encoding-encoding boundary-non-boundary similarity) and across experimental phases (i.e., encoding-recall boundary-non-boundary similarity), including or excluding the two periods. The results confirmed that including or excluding the two non-boundary periods overlapping with within-movie boundary periods does not change any of the pattern similarity measures (two-tailed paired *t*-tests, all *t*(14)s < 1.45, all *p*s >.17). We now report the overlap between non-boundary and within-movie boundary periods and its effect in the Methods section of the revised manuscript as below:

p.21. “Two of the non-boundary periods partially overlapped with the within-movie boundary periods by 13.5 seconds and 4.5 seconds, respectively, and were excluded when correlating within-movie boundary patterns and non-boundary patterns. Note that the two non-boundary periods were included in other analyses in the current study comparing between-movie boundary patterns and non-boundary patterns. However, excluding or including the two non-boundary periods did not significantly change any of the mean pairwise between-movie correlations across (1) encoding non-boundary patterns, (2) encoding non-boundary and between-movie boundary patterns, (3) encoding non-boundary and recall non-boundary patterns, and (4) encoding non-boundary and recall between-movie boundary patterns in PMC (two-tailed paired-samples *t*-tests, all *t*(14)s < 1.45, all *p*s >.17).”

A remaining important question would be whether there are consistent within-movie boundary patterns distinct from non-boundary patterns. To test this, we performed a new analysis: we first computed the mean PMC pattern similarity between within-movie boundaries across different movies, and then compared it against the mean pattern similarity between the within-movie boundary patterns and the non-boundary patterns during encoding across different movies. The two non-boundary patterns that overlapped with within-movie boundaries were excluded from this analysis. We found that there were positive correlations between within-movie event boundary patterns, and the correlation was higher than the correlation between within-movie boundary patterns and non-boundary patterns, as shown in Figure 4—figure supplement 1. In other words, during encoding, within-movie event boundary patterns were more similar to other within-movie event boundary patterns from different movies, than to non-boundary patterns in different movies. Although the overall similarity between within-movie event boundary patterns was much lower than the similarity between movie offset patterns, this may suggest the presence of consistent within-movie boundary patterns not identical to non-boundary patterns.

We revised the Results and Methods sections accordingly:

p.9. “Thus, we hypothesized that boundaries between movies (i.e., between mental contexts) would manifest as stronger versions of within-movie boundaries with qualitatively similar patterns; in other words, boundary patterns would generalize across different scales of boundaries. To test this idea, we first confirmed that there were consistent within-movie event boundary patterns in PMC during encoding; within-movie boundary patterns were more similar to each other than to non-boundary patterns (Figure 4—figure supplement 1).”

pp.20-21. “We first examined whether there were consistent activation patterns following the within-movie event boundaries distinct from non-boundary patterns (Figure 4—figure supplement 1). For each subject, we generated the mean PMC activation pattern for each within-movie boundary by averaging patterns from 4.5 to 19.5 seconds following the within-movie boundary during encoding. We then computed pairwise between-movie Pearson correlations across the within-movie boundary patterns, and averaged the correlations. A two-tailed one-sample *t*-test against zero was performed to test whether the similarity between the within-movie boundary patterns was overall positive. We also computed pairwise between-movie correlations across the within-movie boundary patterns and non-boundary patterns during encoding. The non-boundary pattern for each movie was generated by averaging activation patterns within the middle 15 seconds of the movie (time window shifted forward by 4.5 seconds). A two-tailed paired-samples *t*-test was performed to test whether the similarity between within-movie boundary patterns was greater than the similarity between within-movie boundary patterns and non-boundary patterns.”

d) I was not sure how the "higher-than-average" and "lower-than-average" regions were defined, specifically. What is the average in this case?

The average activation was the average BOLD responses of all volumes collected within a scanning run, and the BOLD responses were z-scored within each run. Thus, the average value was z = 0, and the whole-brain maps in Figure 1 show areas that were relatively activated (positive z) or deactivated (negative z) following between-movie boundaries compared to the average. We have revised the caption for Figure 1 to clarify what the average activation means.

p. 4. “Blue areas indicate regions with lower-than-average activation, where the average activation of a scanning run was z = 0. Likewise, red areas indicate regions with higher-than-average activation.”

e) Page 6, line 130: this phrasing was confusing to me: "this correlation was higher at boundaries than at non-boundaries (Figure 2A, blue arrows)." It made me think that the "blue arrows" in the parentheses may refer to the non-boundaries, but the blue arrows in the figure appear to be across-movie correlation for both boundary (left activation pattern pair in Figure 2A), and non-boundary (right pair Figure 2A) time points. Perhaps rephrase? Maybe "Figure 2A, a versus b blue arrows"?

We apologize for the ambiguous phrasing. We have revised the text as below:

p.5. “We performed a whole-brain pattern similarity analysis on the recall data to identify regions where (1) boundary period activation patterns were positively correlated across different recalled movies (Figure 2A, blue arrow *a* > 0), and (2) this correlation was higher at boundaries than at non-boundaries (Figure 2A, blue arrows *a* > *b*).”

f) Another phrasing question: "…, suggesting that the between-movie boundary pattern may reflect a cognitive state qualitatively different from the state elicited by event boundaries during movie watching (e.g., attentional engagement; Song et al., 2021)." I'm not quite sure what the authors mean by "e.g., attentional engagement". Can you explain further? Differences in attentional engagement across these two experience types, perhaps?

By mentioning the “attentional engagement,” we meant a possible increase in attentional engagement or arousal at within-movie event boundaries. We speculated that this increased attention at event boundaries might be different from the cognitive states associated with the complete reconfiguration of situation models that might occur at between-movie boundaries. We revised the sentence in the Results section as below, and provided further explanation in the Discussion section to clarify this point:

p.9. “These results suggest that the between-movie boundary pattern may reflect a cognitive state qualitatively different from the state elicited by within-movie event boundaries during movie watching.”

p.15. “We speculate that the between-movie boundary state may be a temporary “relay” state that occurs when no one mental model wins the competition to receive full attentional focus following the flushing of the prior mental context. Namely, when one major mental context switches to another, the brain may pass through a transient off-focus (Mittner et al., 2016) or mind-blanking (Mortaheb et al., 2022; Ward and Wegner, 2013) state which is distinct from both processing external stimuli (e.g., movie watching) and engaging in internal thoughts (e.g., memory recall). This account may also explain the difference between within- vs. between-movie boundary patterns: in terms of attentional fluctuation (Jayakumar et al., 2022; Song, Finn, et al., 2021), external attention is enhanced at within-movie event boundaries (Pradhan and Kumar, 2021; Zacks et al., 2007), whereas the relay state is associated with lapses in attention (deBettencourt et al., 2018; Esterman et al., 2014).”

g) As may be clear from my public comment #2, the presentation of results in Figure 3D was hard for me to understand. Perhaps the authors could think of ways to plot this differently or spell out more explicitly the patterns of interest in the paper or legend to help readers understand. I would have loved more hand-holding for this analysis.

We apologize for the unclear description of the offset vs. onset analysis depicted in Figure 3D. To clarify the logic behind the analysis and the results, we now provide more detailed explanations in the Results section of the manuscript as below:

p.7. “Is the generalized boundary pattern evoked by the onset of a movie, rather than the offset? We examined this question by comparing the temporal emergence of the generalized boundary pattern following movie offsets versus onsets (Figure 3D); note that the offset of a movie was temporally separated from the onset of the following movie during both encoding and recall (see Figure 1A). Specifically, we extracted the mean time series of PMC activation patterns around between-movie boundaries, time-locked to either the onset or offset of each watched or recalled movie. We then computed between-phase (encoding-recall) pattern similarity across the individual time points of the activation pattern time series. We found that significantly positive between-phase correlations emerged well before the encoding and recall onsets (Figure 3D, left panel), starting from 4.5 seconds following the offsets of the preceding watched or recalled movie (Figure 3D, right panel). Thus, boundary patterns were not exclusively triggered by movie onsets; it is likely that offset responses significantly contributed to the boundary patterns.”

Following the reviewer’s suggestion, we also indicated the delay duration between onsets and offsets on Figure 3D and Figure 3—figure supplements 2 and 4, by marking the onsets/offsets of the following/preceding movies with dotted lines

Reviewer #2 (Recommendations for the authors):We really enjoyed this manuscript and had very few suggestions or requests for tweaks. We note these below:1. We recognize that this is a short-format submission, but honestly, the 'Discussion' section of this paper just felt far too short. These are very interesting and thought-provoking results, which did not feel commensurate with the very small amount of text devoted to unpacking the findings and their implications. Note that we are not suggesting the authors did a poor job of constructing these two paragraphs, but rather we think that this is simply not enough space to really do justice to a broader conversation that can be had about these findings. We suggest (if possible, from an editorial standpoint) that a lengthier discussion could really solidify the contributions of this paper and make the connections to other studies and other ideas about event cognition and event memory more explicit. In particular, I would love to hear a more fleshed-out exploration of what this general cognitive state that may exist between events might look like, and what it might be doing.

[Copied from Essential Revisions 7] We thank the reviewers for their suggestions. We expanded the discussion on the potential nature of the general boundary state in the revised Discussion section. We also added extended text relating our results to existing empirical and theoretical work. Please see the revised manuscript for the remaining parts of the Discussion section.

2. The authors operationalized boundary periods as "…the first 15 seconds following the offset of each recalled movie, and the non-boundary periods…" as "…the 15 seconds in the middle of each recalled movie." Why was a window of 15 seconds chosen? While I do not have any reason to suspect the results would be wildly different if, say, a 10 or 20 second window was used, we think some justification or indication of why this particular parameter was chosen would be good to include.

[Copied from Essential Revisions 2] Our choice of the boundary/non-boundary period time windows was primarily based on the time courses of boundary-related univariate responses, examined during the basic exploration of recall offset effects. As shown in Figure 1—figure supplement 1, boundary-related signals tended to last up to 30 seconds from recall offset, with the largest responses (deactivation) observed around approximately 15 seconds from the offset. Thus, we decided to use a time window of 15 seconds (shown as a red bar on the x axis) to include most of the time points that showed boundary responses.

In addition, as shown in Figure 3D, the time-time correlations between activation patterns around boundaries indicated that the generalized boundary activation pattern in PMC lasted until around 15 seconds from the movie/recall offset. That is, activation patterns remained stable for at least about 15 seconds from the offset. Thus, we aimed to generate clearer activation patterns to be used for analyses by taking the average of the stable patterns across time points that spanned the 15-s window.

We now clarify our decision for using a relatively long time window of 15 seconds in the Methods section.

3. Regarding Figure 2, we personally think that a different color gradient might be a better choice for the dark gray inflated brain image. This is a minor nit-pick, but the darker red colors do not show up against the dark sulcus portions of the map particularly well, and a different gradient might be preferable. This would be particularly true if individuals were to view this figure in grayscale, or in cases of color-blindness.

We thank the reviewers for their suggestion to use an alternative color gradient so that our manuscript becomes more accessible to colorblind and visually impaired readers. Following the reviewers’ comment, we changed the colormap used in Figure 2B-C.

4. If so many individual parcels from the Schaefer atlas were examined, it is not clear why PCC and Precuneus were combined into a PMC ROI. While we do not particularly suspect that PCC and precuneus should look very different, we think a brief note of why these regions were combined and many others weren't would be good to include.

[Copied from Essential Revisions 4] We created a single PMC ROI mask by combining the PCC and precuneus to be consistent with our prior study that analyzed the same region of interest (Lee and Chen, 2021). Our decision to use PMC including both the PCC and precuneus in our current and prior studies was based on the atlas of functional networks widely used in the field (Schaefer et al., 2018). In the Schaefer atlas, the two regions together form the posterior medial sub-region of the default network, and the parcels that we combined to create the PMC ROI all fell within the default network. In addition, we observed the content- and task-general boundary pattern in the larger PMC area in our whole-brain analysis (Figure 2C), and thus we opted to use a single large PMC mask for our ROI analysis rather than breaking the continuous area into several sub-regions. As Reviewer 2 mentioned in their Recommendations #4, it is unlikely that the PCC and precuneus would show qualitatively different results, because the whole-brain analysis using smaller parcels already demonstrated that the PCC and precuneus as well as their fine-grained sub-areas all showed boundary patterns consistent across experimental phases. We revised the Methods section of the manuscript as below to explain our reasons for combining the two regions:

p.17. “For region-of-interest analyses, we defined the bilateral posterior-medial cortex (PMC) by combining the parcels corresponding to the precuneus and posterior cingulate cortex within Default Network A as in our prior study (Lee and Chen, 2021). The precuneus and posterior cingulate cortex together spanned the area that showed the strongest content-and task-general boundary patterns in the whole-brain analysis (Figure 3C). The bilateral angular gyrus ROI consisted of the parcels corresponding to the inferior parietal cortex within Default Network A, B and C.”

5. Regarding the correlations between the average between-movie boundary pattern and (1) between-movie boundaries, (2) moments of silence, (3) within-movie boundaries (i.e., the data displayed in Figure 4), we had to re-read this figure caption and Results section a few times to feel that we really grasped what comparisons were being made, and how these correlations were being conducted. We think this could stand to be tweaked to be clearer, with perhaps a conceptual diagram like the one in Figure 2A included to aid in understanding.

[Copied from Essential Revisions 9] We apologize for the ambiguity in visualizing our results in Figure 4. To help readers understand the figure, we divided Figure 4 into two different figures: Figure 4 reporting the between-movie vs. within-movie boundary pattern comparison, and Figure 5 reporting control analyses regarding the silence at boundaries (also see our response to Comment #10). In the revised Figure 4, we added a new panel explaining the analysis method with a detailed caption. We used a color scheme that connects the specific comparisons depicted in the analysis schematic (Figure 4A) and the conditions reported in the result graph (Figure 4B). Note that we now report results from PMC only, as the within- vs. between-movie boundary comparison in the auditory cortex was not the focus of the analysis and was not discussed in the text.

We also revised the Results section and the Methods section. We report the correlation between the between-movie and within-movie boundary patterns within the encoding phase (in addition to the cross-phase correlation) to provide more complete results from the analysis.

6. Following from the above point, we were initially not totally clear on the relevance or importance of the 'no sound' condition. Upon reading further into the Results, it made sense, but seeing as this figure is referenced well before this condition is expanded upon, we think this could be motivated a little earlier to mitigate unnecessary confusion. As of now, this set of analyses seems to be motivated after presentation of the data rather than before, and simply shifting this around could aid in understanding.

[Copied from Essential Revisions 10] We thank the reviewers for their suggestion. As we wrote in our response to the comment #5 above, we divided Figure 4 into two so that the revised Figure 4 shows the comparison between within- and between-movie boundary patterns only, and the new Figure 5 shows the control analysis corresponding to the ‘no sound’ condition in the original Figure 4. We now present Figure 5 after the paragraph describing the control analyses, so that there is no confusion due to the order in which the results and figures are presented.

Reviewer #3 (Recommendations for the authors):Specific suggestions the authors related to the points mentioned above:1. To address the effect of similarity between encoding and recall due to the (lack of) perceptual input, it would be relevant to add an additional comparisons to the analyses as shown in figure 2: the similarity between middle segments during recall and boundary segments during encoding.

We thank the reviewer for suggesting this analysis. Following the reviewer’s suggestion, we performed a new whole-brain pattern similarity analysis directly comparing the similarity between encoding and recall boundary patterns against the similarity between encoding boundary patterns and recall non-boundary patterns. As visual input was consistent across the boundary and non-boundary periods during recall, this analysis allowed us to control for the effects of shared visual features on the generalized boundary patterns. [Copied from Essential Revisions 1] That is, if the generalized boundary patterns that we observed were predominantly driven by shared low-level visual features (i.e., subjects viewed mostly-blank screens during boundary periods in both encoding and recall phases), the similarity between encoding and recall boundary patterns would be approximately the same as the similarity between encoding boundary patterns and recall non-boundary patterns. However, we found that several cortical parcels, especially within the DMN, showed significantly greater encoding-recall boundary pattern similarity compared to the similarity between encoding boundary patterns and recall non-boundary patterns; the resulting maps were highly similar to the results reported in Figure 2C. Thus, the consistent boundary patterns observed in higher associative areas were not likely to be mainly due to low-level visual features shared across the experimental phases. We now report this analysis in Figure 2—figure supplement 2.

We revised the Results and Methods sections of the manuscript accordingly.

2. I wonder why the authors choose to focus on a long window of 15 seconds after stimulus onset rather than a shorter window. I would suggest investigating the similarity between the within-movie and between-movie event boundaries by focusing on a shorter window (e.g. 2 TRs). Additionally, I would propose running an analysis like the one in figure 3D for the within-movie and between-movie event boundaries to see how the (dis)similarity develops over time.

[Copied from Essential Revisions 2] Our choice of the boundary/non-boundary period time windows was primarily based on the time courses of boundary-related univariate responses, examined during the basic exploration of recall offset effects. As shown in Figure 1—figure supplement 1, boundary-related signals tended to last up to 30 seconds from recall offset, with the largest responses (deactivation) observed around approximately 15 seconds from the offset. Thus, we decided to use a time window of 15 seconds (shown as a red bar on the x axis) to include most of the time points that showed boundary responses.

In addition, as shown in Figure 3D, the time-time correlations between activation patterns around boundaries indicated that the generalized boundary activation pattern in PMC lasted until around 15 seconds from the movie/recall offset. That is, activation patterns remained stable for at least about 15 seconds from the offset. Thus, we aimed to generate clearer activation patterns to be used for analyses by taking the average of the stable patterns across time points that spanned the 15-s window.

We now clarify our decision for using a relatively long time window of 15 seconds in the Methods section.

3. It is very unclear at this point whether the observed pattern of neural responses is related to the specific setup of the study or whether subtle shifts in context versus more intense transitions involve fundamentally different mechanism. I understand that this question probably cannot be answered with the current dataset, but I would appreciate some additional discussion related to this question, as well suggestions for how this could be clarified in future studies. I think the paper might be more suitable for a longer format, with a separate results and Discussion section. That would allow for a more extensive discussion for what the results might mean for our current understanding of how event segmentation works.

[Copied from Essential Revisions 6, 8] We agree with the reviewer that the manuscript can be improved by separating the Results and Discussion sections and adding more in-depth treatment of the findings. Following the reviewer’ suggestions, we divided Results and Discussion into two separate sections and substantially expanded the Discussion. We now provide more in-depth discussion on the within- vs. between-movie boundary distinction (i.e., subtle vs. intense transitions) in the revised Discussion section as below:

p.14. “Although the boundary-related PMC activation patterns were consistent across internally- and externally-driven boundaries, they did not generalize across within- and between-movie boundaries. Relatedly, a recent human neurophysiological study (Zheng et al., 2022) reported that medial temporal cortex neurons distinguished within- and between-movie boundaries while subjects were watching short video clips; some neurons responded only to between-movie boundaries, whereas a separate group of neurons responded to both types of boundaries. These findings may be in line with the view that event boundaries have a hierarchical structure, with different brain areas along the information pathway reflecting different levels of boundaries, from fine-grained sensory transitions to coarse-grained situational transitions (Baldassano et al., 2017; Chang et al., 2021; Geerligs et al., 2021). However, it is still puzzling that within- and between-movie boundaries in our study produced qualitatively distinct neural patterns within a highest-order area (PMC), even though both categories consisted of prominent boundaries between situations spanning tens of seconds to several minutes. What are the crucial differences between the two levels of boundaries? One important factor might be the presence or absence of inter-event connections. Even the most salient within-movie boundaries still demand some integration of information across events, as the events are semantically or causally related, and ultimately constitute a single coherent narrative (Lee and Chen, 2021; Song, Park, et al., 2021). In contrast, an entire cluster of related events, or the narrative as a whole, might be completely “flushed” at between-movie boundaries; this difference could induce distinct cognitive states at the two levels of boundaries, giving rise to different PMC patterns.”

In addition, in the last paragraph of the Discussion section, we suggest a future study for testing whether our findings would be generalized in more realistic and unconstrained settings:

pp.15-16. “In conclusion, we found that internally-driven boundaries between memories produce a stereotyped activation pattern in the DMN, potentially reflecting a unique cognitive state associated with the flushing and updating of mental contexts. By demonstrating stimulus-independent event segmentation during continuous and naturalistic recall, our study bridges the gap between the fields of event segmentation and spontaneous internal thoughts (also see Tseng and Poppenk, 2020). Without any task demands or external constraints, the mind constantly shifts between different internal contexts (Raffaelli et al., 2021; Sripada and Taxali, 2020). What are the characteristics of neural responses to different types of spontaneous mental context boundaries (e.g., between two different memories, between external attention and future thinking)? Is the boundary pattern observed in the current study further generalizable to mental context transitions even more stark than between-movie transitions in our experiment? Are there specific neural signatures that predict subsequent thought transitions? Future work will explore answers to these questions by employing neuroimaging methods with behavioral paradigms that explicitly and continuously track the unconstrained flow of thoughts in naturalistic settings.”